# Archaean multi-stage magmatic underplating drove formation of continental nuclei in the North China Craton

Jin Liu[1,2], Richard M. Palin [2] ✉, Ross N. Mitchell [3,4], Zhenghong Liu[1,5], Jian Zhang[6], Zhongshui Li[7], Changquan Cheng[8] & Hongxiang Zhang[1]

The geodynamic processes that formed Earth's earliest continents are intensely debated. Particularly, the transformation from ancient crustal nuclei into mature Archaean cratons is unclear, primarily owing to the paucity of well-preserved Eoarchaean–Palaeoarchaean 'protocrust'. Here, we report a newly identified Palaeoarchaean continental fragment–the Baishanhu nucleus–in northeastern North China Craton. U–Pb geochronology shows that this nucleus preserves five major magmatic events during 3.6–2.5 Ga. Geochemistry and zircon Lu–Hf isotopes reveal ancient 4.2–3.8 Ga mantle extraction ages, as well as later intraplate crustal reworking. Crustal architecture and zircon Hf–O isotopes indicate that proto-North China first formed in a stagnant/squishy lid geodynamic regime characterised by plume-related magmatic underplating. Such cratonic growth and maturation were prerequisites for the emergence of plate tectonics. Finally, these data suggest that North China was part of the Sclavia supercraton and that the Archaean onset of subduction occurred asynchronously worldwide.

Earth is the only known planet to have evolved a felsic continental crust. While the majority of continental crustal growth occurs today at convergent plate margins, the dominant crust-forming mechanisms that operated during the Archaean (and even Hadean) are strongly disputed[1]. Some authors argue that subduction and plate tectonics has operated on Earth since at least the Eoarchaean[2,3]. By contrast, numerical modelling and field investigation of some Palaeoarchaean cratons suggests that a 'stagnant lid' regime operated, where lithospheric plates moved very slowly across Earth's surface–if at all. In such an environment, the formation of felsic/TTG-like crust was driven by mantle plume activity[4] and/or melting within lithospheric drips[5,6]. This latter geodynamic scenario finds support from the study of Mars and Venus, which can be considered

analogues for the early Earth[7]. Determining when, where, and why certain geodynamic regimes dominated at different points in time has key implications for other critical events in Earth history, such as continental emergence, atmospheric oxygenation, changes in ocean composition, and the appearance and evolution of life[8]. The primarily obstacle in resolving this dispute is the scarcity of preserved Hadean to early Archaean continental crust on the modern Earth. Even in cratons that contain such Eoarchaean rocks, they often only comprise a volumetrically minor component of the terrane itself[9]. Therefore, any newly discovered ancient nuclei can provide invaluable insights into the crustal evolution of early Earth, and how cratons grew and eventually matured. The North China Craton preserves a continuous record of successive magmatic events that

[1]College of Earth Sciences, Jilin University, Changchun, China. [2]Department of Earth Sciences, University of Oxford, Oxford, UK. [3]State Key Laboratory of Lithospheric Evolution, Institute of Geology and Geophysics, Chinese Academy of Sciences, Beijing, China. [4]College of Earth and Planetary Sciences, University of Chinese Academy of Sciences, Beijing, China. [5]Key Laboratory of Mineral Resources Evaluation in Northeast Asia, Ministry of Natural Resources, Changchun, China. [6]Department of Earth Sciences, The University of Hong Kong, Hong Kong, China. [7]College of Exploration and Geomatics Engineering, Changchun Institute of Technology, Changchun, China. [8]School of Earth Sciences and Engineering, Sun Yat-sen University, Zhuhai, China. ✉ e-mail: richard.palin@earth.ox.ac.uk

span almost all (3.8–2.5 Ga) of the Archaean Eon[10]. Two major continental nuclei have been previously established: the Anshan continental nucleus (in the Anshan–Benxi areas) and the Eastern Hebei continental nucleus (Fig. 1a), both of which have ancient geologic histories dating back to as old as 3.8 Ga (ref. 10). The region stretching from Anshan to Jiapigou (Fig. 1b) preserves the most comprehensive record of Archaean magmatism within North China and represents an ideal natural laboratory for investigating the formation and evolution of an Archaean craton.

Here, we report a Palaeoarchaean continental nucleus, named the Baishanhu nucleus (occurring in the Jiapigou–Baishanhu areas), which has been identified in northeastern North China Craton. This study presents zircon U–Pb geochronology from a suite of granitoids, alongside zircon Lu–Hf and O isotopic analyses and whole-rock geochemistry. These results establish a spatiotemporal framework for the formation of the Baishanhu nucleus, which provides constraints on cratonic growth and maturation, and shed light on the geodynamic regime that likely operated on Earth during the early Archaean.

## Results and discussion
### Geochronological framework
To establish craton growth and maturation processes of the Baishanhu nucleus, we conducted a total of 870 zircon U–Pb geochronological analyses from 21 samples. Detailed description of zircon U–Pb dating, as well as newly acquired and compiled zircon Hf–O data, can be found in the Supplementary Notes and Supplementary Data 1–6. The results reveal at least five discrete magmatic episodes at 3.6–3.5, 3.3–3.2, 2.8–2.7, 2.63, and 2.55–2.50 Ga (Fig. 2a). An early Palaeoarchaean magmatic event is recorded by abundant 3.6–3.5 Ga xenocrystic zircons found within younger (3.3–3.2 and 2.55–2.50 Ga) granitoids. These xenocrysts exhibit concentric oscillatory internal zonation, suggestive of crystallisation from a felsic melt (Fig. 2b). Evidence for a subsequent late Palaeoarchaean/early Mesoarchaean magmatic event comes from 3.3–3.2 Ga monzogranites and 3.3–3.2 Ga xenocrystic zircons within younger 2.55–2.50 Ga potassic granitoids. These 3.3–3.2 Ga monzogranites are the oldest potassic granites documented within the North China Craton, and primarily consists of plagioclase (35%), microcline (25%), quartz (30%), biotite (5%), and minor hornblende. An

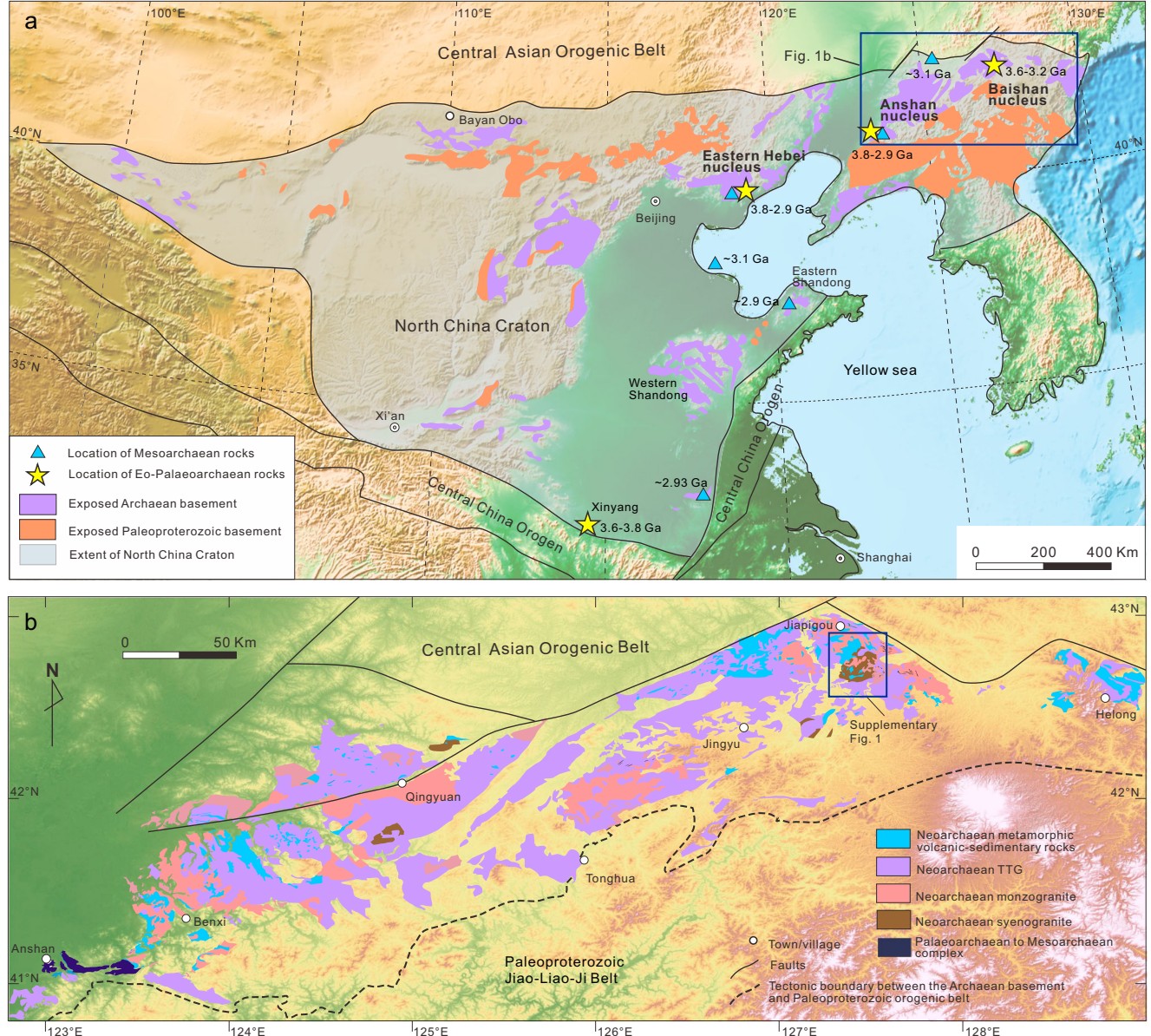

**Fig. 1 | Geological map of North China Craton and the area studied. a** Map showing outcrops of Archaean–Paleoproterozoic basement and reported pre-Neoarchaean rocks in North China Craton (modified from Wan et al.[1]); **b** Geological map from Anshan to Jiapigou (modified from Guo et al.[2]).

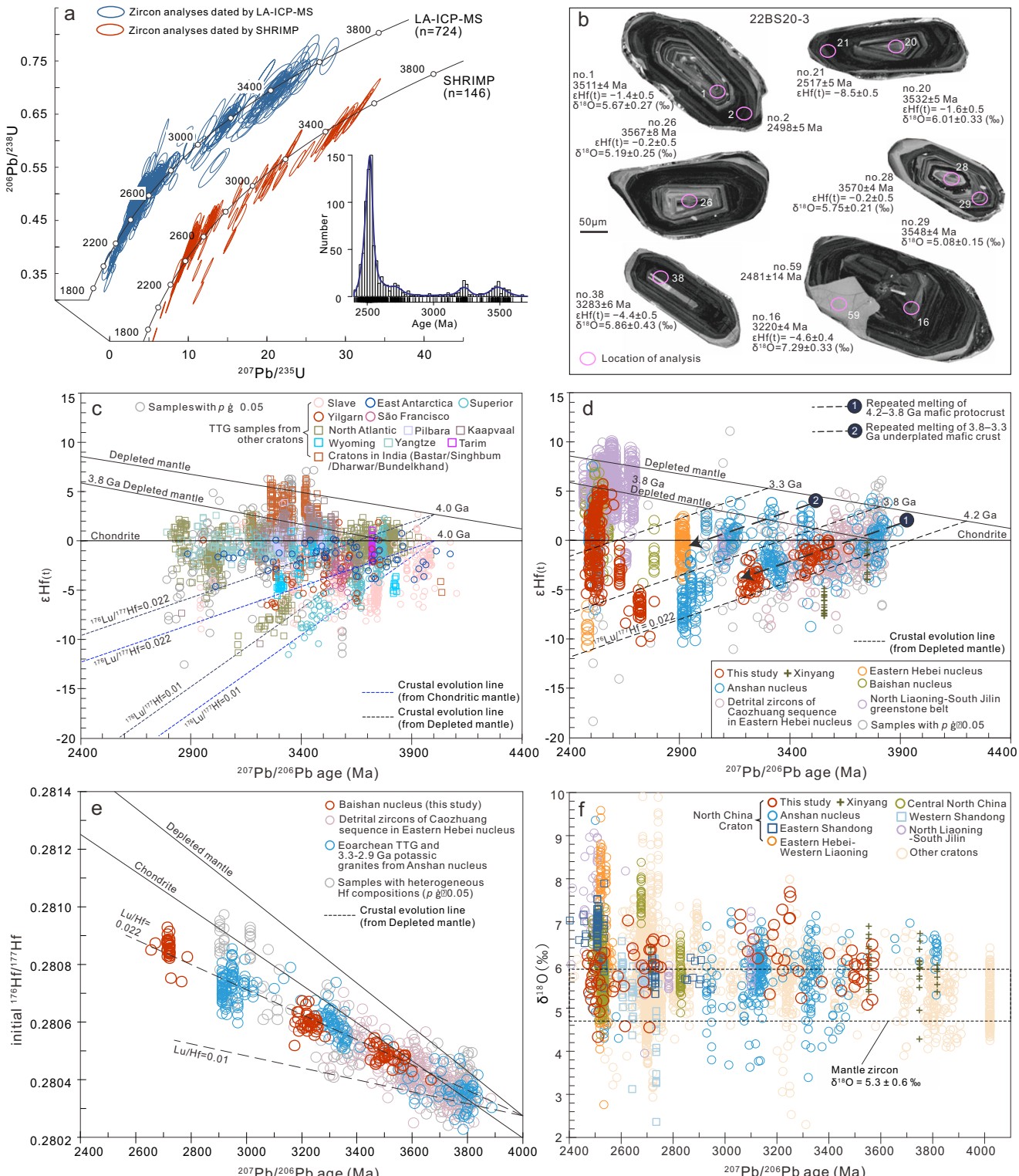

**Fig. 2 | Zircon U–Pb dating and Hf–O isotopes. a** Zircon concordia diagram and kernel density estimate plot for Palaeoarchaean–Neoarchaean granitoids. **b** Representative zircon CL images of the c. 3.3 Ga monzogranite sample 22BS20-3. **c** Compilation of zircon epsilon hafnium (εHf(t)) values of Eoarchaean-to-Mesoarchaean TTGs from other ancient cratons versus ²⁰⁷Pb/²⁰⁶Pb age. **d** Compilation of zircon εHf(t) values from the North China Craton versus ²⁰⁷Pb/²⁰⁶Pb age. **e** Diagram of initial ¹⁷⁶Hf/¹⁷⁷Hf value versus ²⁰⁷Pb/²⁰⁶Pb age.

**f** Compilation of zircon δ¹⁸O values versus ²⁰⁷Pb/²⁰⁶Pb age. Note only analyses showing no significant Pb loss are plotted. Depleted mantle lines are based on models of a present depleted mantle with a ¹⁷⁶Hf/¹⁷⁷Hf of 0.283251 and a ¹⁷⁶Lu/¹⁷⁷Hf of 0.0384 (ref. 68), and the 3.8 Ga Depleted mantle curve is extrapolated from the assumption that growth of the depleted mantle began at 3.8 Ga (εHf = 0) and evolved to the present-day depleted mantle reservoir[14]. The δ¹⁸O value range for "mantle zircon" is from ref. 55.

early Neoarchaean magmatic event (2.8–2.7 Ga) is evidenced by c. 2.78 Ga trondhjemitic gneiss from the Jiapigou area[11], c. 2.72 Ga monzogranite, and 2.8–2.7 Ga xenocrystic zircons within 2.55–2.50 Ga potassic granitoids. The c. 2.72 Ga monzogranite exhibits a massive structure and has a mineral assemblage of quartz (30%), plagioclase (30%), microcline (35%), and minor biotite (5%). Finally, two episodes of late Neoarchaean magmatism within the Baishanhu nucleus are documented by minor c. 2.63 Ga magmatism (monzogranite), and intensive and widespread magmatism at 2.55–2.50 Ga, similar to other terranes within North China. The dominant lithologies of this young c. 2.5 Ga episode include TTG, meta-mafic volcanic rocks, and potassic granitoids.

## Crustal evolution of Baishanhu nucleus revealed by zircon Hf isotopes

The zircon Lu–Hf isotopic system is a useful and robust tool for deciphering the evolutionary history of the continental crust[12]. Here, we acquired 321 new zircon Hf isotope analyses for the Baishanhu nucleus (Fig. 2b). Generally, the two-stage depleted mantle model ($T_{DM}^2$) age of granitoids represents the timing of extraction of their mafic precursors from a depleted mantle source, which in turn signifies the timing of crustal growth[6]; however, we note that the Hf $T_{DM}^2$ age is affected by the assumed mantle depletion history and the $^{176}Lu/^{177}Hf$ ratio of the crust. Currently, the community's understanding of the evolution of the Earth's depleted mantle through time is uncertain. Several studies of Eoarchaean TTGs (e.g., the Itsaq gneiss, North Atlantic Craton[13], and the Aktash gneiss, Tarim Craton[14]) have indicated that their mafic precursors were derived from a near-chondritic mantle. Further, Hf–Nd isotopes of Palaeoarchaean (c. 3.6 Ga) mafic to ultramafic rocks from the Pilbara Craton also indicate the existence of a chondritic to near-chondritic mantle at that time[15,16]. Thus, these authors interpret that any global depletion of the mantle had not begun during the Eoarchaean or Hadean, such that a depleted mantle signature began to progressively develop since c. 3.9 Ga (ref. 16), although see ref. 17 for the potential caveats involved when zircon is used to estimate mantle depletion.

To directly address this issue, we compiled a global dataset of igneous zircon Hf isotopes of early Archaean TTGs from all major cratons on Earth. The compilation reveals that most zircons display very low sub-chondritic $\varepsilon Hf_{(t)}$ values (Fig. 2c), indicating that the mafic precursors of these early Archaean TTGs likely had a long crustal residence time. These unradiogenic Hf isotopes indicate that some crust-mantle differentiation must have occurred in the Hadean to early Archaean, and the inferred source ages for these unradiogenic zircons vary depending on the assumed degree of mantle depletion. Several Palaeoarchaean mafic–komatiitic (e.g., Barberton Greenstone Belt[18] and Western Pilbara Craton[19]) and TTG rocks (e.g., Dharwar Craton[20] and Singhbum Craton[21]) that display depleted mantle signatures also argue for the existence of a depleted mantle in the early Archaean. After excluding analyses that show clear Pb loss and samples with heterogeneous Hf isotopic compositions, many Eoarchaean zircons from North China still record highly depleted Hf isotope signatures (Fig. 2d). Measured $^{142}Nd–^{143}Nd$ isotopes of c. 3.8–3.0 Ga rocks from the Anshan Complex suggest multiple mantle-crust differentiation events between 4.3 Ga and 3.8 Ga, and indicate the existence of a depleted upper mantle during this period[22]. Thus, we interpret that the early Archaean mantle was heterogeneous with some domains (e.g., North China and Dharwar Cratons) that exhibited depleted Hf isotopes, whereas other domains (e.g., North Atlantic and Tarim Cratons) exhibited chondritic compositions. Therefore, the $T_{DM}^2$ age calculation performed in this study assumes that early Archaean mafic protocrust within North China formed from a depleted mantle. Additionally, a plot of zircon ages against their initial $^{176}Hf/^{177}Hf$ ratios demonstrates that the 3.6–2.7 Ga zircon from the Baishanhu nucleus, the 3.8–2.9 Ga zircon from the Anshan nucleus, and the

Eoarchaean–Palaeoarchaean detrital zircon of the Caozhuang sequence from the Eastern Hebei nucleus all fall along a crustal evolution line with a $^{176}Lu/^{177}Hf$ ratio of 0.022 (Fig. 2e). This trend is consistent with the $^{176}Lu/^{177}Hf$ ratio observed in Archaean mafic crust[23]. Thus, we consider it reasonable to use a $^{176}Lu/^{177}Hf$ ratio of 0.022 for calculating the $T_{DM}^2$ ages of the Archaean zircon from North China.

The 3.6–3.5 Ga zircon grains from the Baishanhu nucleus have sub-chondritic $\varepsilon Hf_{(t)}$ values ranging from −3.6 to −0.1, with $T_{DM}^2$ ages of 4.2–3.9 Ga. In contrast, the 3.3–3.2 Ga and c. 2.72 Ga groups display more unradiogenic Hf isotopic features with sub-chondritic $\varepsilon Hf_{(t)}$ values ranging from −6.0 to −1.5 and −10.2 to −4.2, and both groups have similar $T_{DM}^2$ ages of 4.2–3.8 Ga. Zircons from the c. 2.63 Ga monzogranite exhibit sub-chondritic $\varepsilon Hf_{(t)}$ values ranging from −4.5 to −0.5 with $T_{DM}^2$ ages of 3.7–3.3 Ga. Some of the 2.55–2.50 Ga potassic granites also exhibit unradiogenic Hf isotopes, yielding sub-chondritic $\varepsilon Hf_{(t)}$ values ranging from −7.7 to −1.1, with $T_{DM}^2$ ages of 3.7–3.3 Ga. In contrast, several 2.55–2.50 Ga potassic granites have more radiogenic Hf isotopes with positive $\varepsilon Hf_{(t)}$ values ranging from 0 to +5.7 with $T_{DM}^2$ ages of 3.2–2.8 Ga. Together, these data show that the 3.6–3.5, 3.3–3.2, and c. 2.72 Ga zircons have a similar $T_{DM}^2$ age range (4.2–3.8 Ga), and they also display a common evolution as demonstrated by well-defined 4.2 and 3.8 Ga crustal evolution lines (Fig. 2d). The 4.2–3.8 Ga $T_{DM}^2$ ages thus represent the mantle extraction age of the mafic protocrust, which subsequently experienced multiple stages of recycling at 3.6–3.5, 3.3–3.2, and c. 2.72 Ga. In addition, the Hf isotopes of the c. 2.78 Ga trondhjemitic gneiss[11], the c. 2.63 Ga monzogranite, and the 2.55–2.50 Ga potassic granites with sub-chondritic $\varepsilon Hf_{(t)}$ values reflect another mantle extraction event that occurred from 3.7–3.3 Ga (Fig. 2d). Moreover, the 2.55–2.50 Ga potassic granites with radiogenic Hf compositions, as well as the previously reported 2.55–2.50 Ga mafic volcanic rocks and TTGs from the Baishanhu nucleus (refs in Supplementary Data 4), suggest significant juvenile crustal growth between the Mesoarchaean and Neoarchaean. Together, these data demonstrate that the Baishanhu nucleus experienced multiple phases of crustal growth and reworking/recycling processes throughout the Archaean, leading to its geochemical maturation and facilitation of cratonization. Similar Archaean crusts with secular evolved Hf isotopes are also developed in other cratons, such as the Yilgarn and Slave[24].

## Geochemical constraints on petrogenesis of the 3.3–2.5 Ga potassic granites

In this study, a total of 20 samples were analyzed for major and trace elements. Detailed results of geochemistry can be found in Supplementary Data 7. The 3.3–2.5 Ga potassic granites exhibit similar geochemical features, such as high $SiO_2$ (70.38–77.94 wt. %), low MgO (0.18–1.21 wt. %), $^TFe_2O_3$ (1.24–4.00 wt. %), Cr (3.16–26.4 ppm), and Ni (1.22–17.20 ppm) contents. They are enriched in light rare earth elements (e.g., La and Ce), Sr, Zr, and Hf, but depleted in heavy rare earth elements (e.g., Lu, Yb, and Y), Nb, and Ta (Fig. 3a), with mainly positive Eu anomalies. On a $Al_2O_3/(FeO^T+MgO)–(3CaO)–(5K_2O/Na_2O)$ ternary diagram[25] (Fig. 3b), the 3.3–3.2 Ga and c. 2.72 Ga monzogranites plot mainly within the field of melts derived from tonalite. As the 3.3–3.2 Ga and c. 2.72 Ga zircons exhibit a similar Hf crustal evolutionary pattern to the 3.6–3.5 Ga xenocrystic zircons, it is reasonable to propose that the 3.6–3.5 Ga TTG was the source for the 3.3–3.2 Ga and c. 2.72 Ga monzogranites. The source rock for the c. 2.63 Ga monzogranite could have been a high-K mafic rock (Fig. 3b), which had a mantle extraction age of 3.7–3.3;Ga. The c. 2.5 Ga potassic granites exhibiting negative $\varepsilon Hf_{(t)}$ values might have been sourced from TTG rocks (Fig. 3b), whereas the other c. 2.5 Ga potassic granites displaying positive $\varepsilon Hf_{(t)}$ values, higher $Al_2O_3$ values and $K_2O/Na_2O$ ratios, likely formed from juvenile metasediments. All of the 3.3–2.5 Ga potassic granites display high Sr/Y ratios (32–132, except for one analysis of 18) and $La_N/Yb_N$ (35–201, except for one analysis of 17), and can thus be classified as adakitic granites[26] (Fig. 3c). Generally, such adakitic geochemical

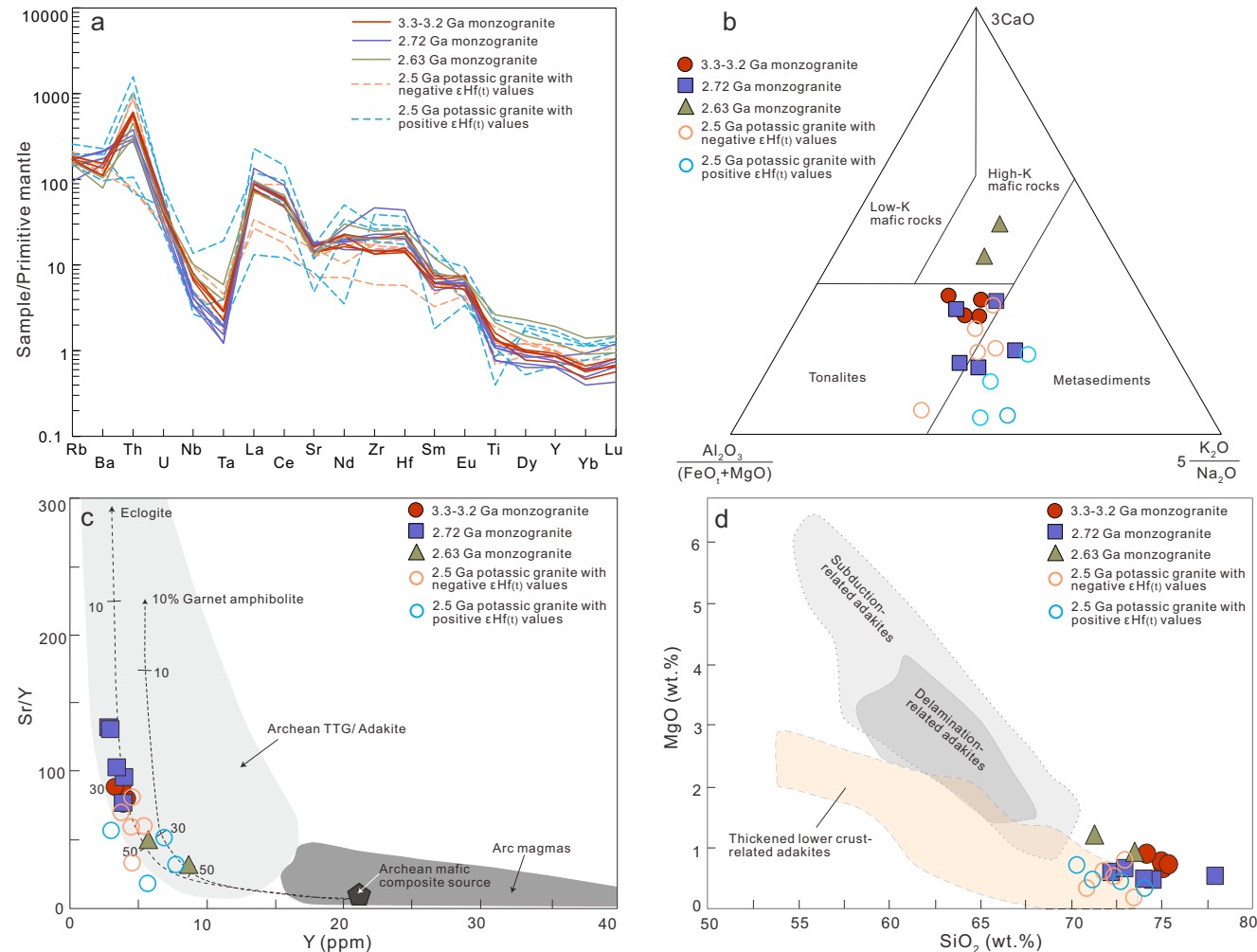

**Fig. 3 | Geochemical diagrams of the 3.3–2.5 Ga granitoids of the Baishanhu nucleus. a** Primitive mantle-normalised multi-element diagram, where normalised primitive mantle values are from Sun and McDonough[69]. **b** $Al_2O_3/(FeO^T +$ MgO)–3*CaO–5*$(K_2O/Na_2O)$ ternary diagram[25]. **c** Sr/Y versus Y diagram. Dashed lines represent basalt partial melting curves leaving either 10% garnet amphibolite or eclogite restite assemblages[26]. **d** MgO versus $SiO_2$ diagram for adakite[28].

characteristics suggest substantial garnet but minor plagioclase in the source region during partial melting, classically interpreted to occur at a pressure greater than 1.0 GPa (ref. 26). Based on the computational method of ref. 27, the average Zr saturation temperatures ($T_{Zr}$) of the 3.3–3.2, c. 2.72, c. 2.63, and c. 2.5 Ga adakitic granites are 794 °C, 847 °C, 818 °C, and 808 °C, respectively. Moreover, the characteristics of high $SiO_2$ contents but low MgO, Cr, and Ni contents suggest a thickened crust origin for these adakitic granites[28] (Fig. 3d). This interpretation is also supported by abundant xenocrystic zircons within the 3.3–2.5 Ga potassic granites and their unradiogenic zircon Hf isotopes (e.g., negative zircon $\varepsilon Hf_{(t)}$ values and Hadean to Palaeoarchaean $T_{DM}^2$ ages). As such, these adakitic granites indicate that the Baishanhu nucleus maintained a notably thick continental crust (>30 km) from at least 3.3 Ga to 2.5 Ga.

**Crustal architecture of northeastern North China and its affinity with other nuclei of the North China Craton**

To contextualise our new data, we present contour maps of zircon U–Pb ages and Lu–Hf data derived from Archaean igneous rocks spanning the region from Anshan to Jiapigou, which illustrate the architecture of the early Archaean basement of the North China Craton (Fig. 4). Although the oldest magmatic record in the Baishanhu nucleus is slightly younger than that in the Anshan nucleus (Fig. 4a), both continental nuclei appear to have initially originated from Hadean to

early Eoarchaean protocrust (4.2–3.8 Ga) (Fig. 4b). An $\varepsilon Hf_{(t)}$ contour map (Fig. 4c) also supports the interpretation that the Anshan and Baishanhu nuclei are dominated by reworked ancient crust, as they show strongly negative $\varepsilon Hf_{(t)}$ values. Additionally, two small regions in the Qingyuan and Helong areas display ancient $T_{DM}^2$ ages of 3.8–3.3 Ga and unradiogenic Hf isotopes, probably indicating the existence of ancient crustal fragments (Fig. 4). This interpretation is further supported by the identification of the c. 3.1 Ga TTG and amphibolite assemblage[29] and abundant 2.9–2.7 Ga xenocrystic zircons within the c. 2.7 Ga meta-mafic volcanic rocks[30]. The Baishanhu nucleus and the other two ancient crustal fragments occur as isolated fragments on the northern margin of the North China Craton, and are separated from the Anshan nucleus and each other by the late Neoarchaean North Liaoning to South Jilin granite–greenstone belt (Fig. 4). This crustal architecture suggests that these ancient nuclei are surrounded by a younger and juvenile granite–greenstone belt. Then, the tectono-thermal event that transpired during the late Neoarchaean (2.7–2.5 Ga) facilitated a significant expansion and growth of the North China Craton, extending it from its original Eoarchaean-to-Mesoarchaean cratonic nuclei.

The three nuclei within the North China Craton (i.e., Anshan, Baishanhu, and Eastern Hebei) are connected by Neoarchaean granite–greenstone belts; however, it remains unclear whether these nuclei belonged to a single coherent continental terrane or

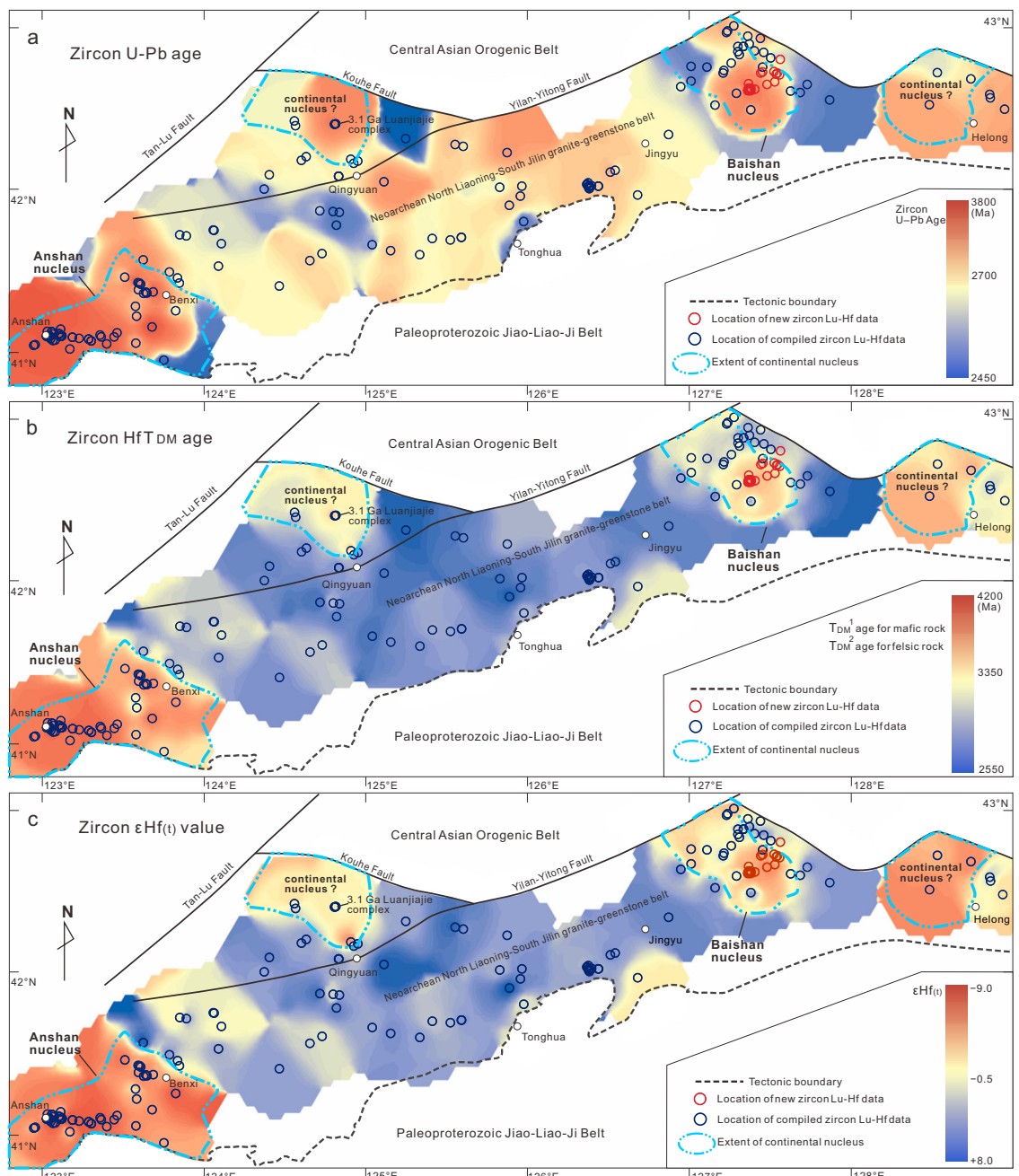

**Fig. 4 | Archaean crustal architecture of the north Liaoning–south Jilin region, North China Craton.** Contour maps of zircon U–Pb age (**a**), zircon Hf T$_{DM}$ age (**b**), and zircon εHf$_{(t)}$ value (**c**), constructed following ref. 70.

represented three individual terranes. In favour of the former argument, we note that these nuclei experienced many similar Eoarchaean-to-Mesoarchaean magmatic events, such as the c. 3.8 Ga magmatic event recorded in the Anshan and Eastern Hebei nuclei[10]. The oldest c. 3.6 Ga magmatic event in the Baishanhu nucleus also correlates with that identified from the Hujiamiao Complex of the Anshan nucleus[10]. Also, intensive crustal reworking events occurred in all three nuclei during 3.3–2.9 Ga (refs. 31–33), leading to the formation of large-scale potassic granitoids and reflecting the existence of voluminous continental crust in each region during the Mesoarchaean. Finally, zircon Hf isotopes suggest similar T$_{DM}^{2}$ ages (4.2–3.8 Ga), $^{176}$Lu/$^{177}$Hf ratios (0.022), and crustal evolution histories (Fig. 2d) for these three nuclei. Therefore, the three North China nuclei may have once constituted a coherent Eoarchaean-to-Mesoarchaean proto-craton. Considering that c. 2.7–2.6 Ga potassic granites have only been identified from the

Baishanhu nucleus, and the existence of a 2.7–2.6 Ga granite-greenstone belt between the Baishanhu and Anshan nuclei, we suggest that Baishanhu might have rifted away from Anshan during the early Neoarchaean, and then later reunited during late Neoarchaean cratonization of North China. Such a scenario involving rifting and breakup of an ancient continental nucleus resembles the processes that have been proposed for the Yilgarn Craton[6].

**Hadean to early Eoarchaean mafic protocrust in early Earth**
As shown by our new data, zircon Hf isotopes indicate that the Anshan, Baishanhu, and Eastern Hebei nuclei originated from a Hadean to early Eoarchaean mafic protocrust (Fig. 2d). $^{142}$Nd and $^{143}$Nd isotopes for the 3.8–3.0 Ga Anshan Complex suggest 4.5–4.4 Ga model ages for the precursor of the oldest components and multiple mantle-crust differentiation events from 4.3 to 3.8 Ga (ref. 22). Several Hadean detrital

zircons[10] and xenocrystic zircons[34] identified from the Eastern Hebei and Anshan nuclei and the southern margin of North China provide direct evidence of a Hadean to early Eoarchaean heritage for the craton. This interpretation is further corroborated by the Xinyang Eoarchaean xenoliths from southern North China[35] that display more evolved Hf isotopes than the other three northern continental nuclei (Fig. 2d). Our new compilation of zircon Hf isotopes of Eoarchaean-to-Mesoarchaean TTGs worldwide show that the vast majority of Eoarchaean–Palaeoarchaean TTGs from most cratons (e.g., Slave, Yilgarn, East Antarctica, Superior, and Kaapvaal) generally display $T_{DM}^2$ ages >4 Ga (Fig. 2c). Studies of the Eoarchaean Acasta Gneiss Complex of the Slave Craton suggest that these oldest rocks on Earth were generated from a Hadean mafic protocrust[9,36]. In light of these earliest records and our new findings from North China Craton, we propose the existence of a Hadean to early Eoarchaean mafic protocrust that was crucial for the formation of the earliest continental crust nuclei preserved in individual cratons[37].

### Repeated Archaean underplating in a plume-dominant regime

Our new data show that the cores of the Baishanhu and Anshan nuclei experienced multiple stages of crustal reworking since the Eoarchaean. The 3.8–3.6 Ga TTG and the c. 3.3 and 3.1–2.9 Ga potassic granites in the Anshan nucleus, as well as the 3.3–2.7 Ga granitoids within the Baishanhu nucleus all exhibit consistent linear crustal evolutionary trends (melting trend #1 in Fig. 2d), and were therefore derived from repeated melting of a 4.2–3.8 Ga mafic precursor(s), without significant addition of juvenile materials in any magmatic episode. In addition, the c. 3.45 Ga migmatite, the c. 3.3 Ga trondhjemite, and the c. 3.1 Ga trondhjemite from the Anshan nucleus, the c. 2.9 Ga TTG and diorite from the Eastern Hebei nucleus, as well as the c. 2.78 Ga trodhjemite, c. 2.63 monzogranite, and some 2.5 Ga potassic granites from the Baishanhu nucleus define another reworking episode with a distinct crustal evolution array (melting trend #2 in Fig. 2d). They represent repeated melting products of former (3.8–3.3 Ga) underplated mafic crusts. This tectono-magmatic history of repeated intervals of the reworking of ancient continental crust best fits a scenario involving multiple underplating episodes within a plume-dominated environment, most parsimoniously characteristic of a stagnant lid geodynamic regime. The alternative scenario involving crustal growth and reworking driven by arc magmatism above a subduction zone would typically involve the incorporation of more juvenile components with radiogenic Hf isotopes, which is not observed until c. 2.6 Ga. Furthermore, the lithospheric architecture, characterised by younger and juvenile granite–greenstone belts surrounding ancient, long-lived, and reworked continental nuclei (Fig. 4), notably differs from linear arcs and collisional orogens typically associated with Phanerozoic convergent plate margins[6].

Our data, expanding on previous interpretations, allow us to propose a tectonic model for the North China Craton (Fig. 5). (i) The Anshan and Baishanhu nuclei initially formed in a stagnant lid environment in an oceanic plateau setting due to partial melting of a 4.2–3.8 Ga mafic protocrust at 3.8 Ga and 3.6 Ga, respectively (Fig. 5a). (ii) Underplating of mafic magmas during the late Palaeoarchaean to Mesoarchaean drove reworking of the existing Anshan and Baishanhu Eoarchaean–Palaeoarchaean continental crusts, causing crustal anatexis and the production of voluminous 3.3–2.9 Ga potassic granites (Fig. 5b). (iii) A final stage of magmatic underplating would have driven further recycling of Eoarchaean–Palaeoarchaean continental crust in the Baishanhu nucleus and the generation of c. 2.72 and 2.63 Ga monzogranites, and caused the Baishanhu and Anshan nuclei to separate (Fig. 5c). Similar processes also dominated the growth and maturation of other ancient cratons. For instance, the 3.5–2.8 Ga granitoids from the East Pilbara Craton display a secular trend towards more evolved Hf isotopes, transitioning from TTG to K-rich granite with limited addition of juvenile materials under a plateau-type setting[38]; in the Yilgarn Craton, several episodes of crustal reworking events occurred during 3.7–2.8 Ga, driven by mantle plume activities[6,39]; multiple episodes of underplating from 3.6 to 3.2 Ga were also recognised from the Kaapvaal Craton[40]. Therefore, we propose that the growth and maturation of Archaean cratons required multiple cycles of underplating under a vertically-dominated, plume-related regime. Such mantle processes also resulted in the stabilisation of thick subcontinental lithospheric mantle roots. Stabilisation of such cratonic nuclei and the appearance of more rigid, plate-like crustal fragments was likely the tipping point for allowing the global transition to plate tectonics and the onset of the supercontinent cycle[41].

### Archaean supercratons and the asynchronous onset of subduction

It has been proposed that the presently separated Archaean cratons might have once constituted a larger ancestral landmass(es), either a single supercontinent ("Kenorland")[42] or else several individual supercratons[43], during the late Archaean. A recent paleomagnetic study performed on the c. 2.62 Ga Yandinilling dike swarm of the Yilgarn Craton in western Australia supports the hypothesis of multiple, long-lived supercratons having existed through the Archaean–Proterozoic transition[44]. Three or so supercratons, such as Superia, Sclavia, and Vaalbara, have been proposed based on their distinct histories of amalgamation and breakup[43]. Some plate reconfiguration schemes suggest the Superia supercraton, including the Superior, Hearne, and Kola/Karelia cratons, and the Vaalbara supercraton, mainly consisting of the Kaapvaal and Pilbara cratons, were joined together as a larger Supervaalbara supercraton[45]. Separately, the Sclavia supercraton is thought to have included the Slave (Canada), Yilgarn (Australia), Zimbabwe, Dharwar (India), and São Francisco (Brazil) cratons[43,44,46,47]. Within these supercraton reconstructions, North China has been conspicuously absent; however, several key geological similarities lead us to argue that North China appears to have a close affinity with the Sclavia supercraton, consistent with recent speculation[48]. For example, (i) similar Hadean–Eoarchaean tectono-thermal events, as evidenced by U–Pb ages and Hf isotopes of igneous zircons (Fig. 2c, d) from the Acasta complex in the Slave Craton[9,12], the Narryer terrane in the Yilgarn Craton[39], the Anshan and Eastern Hebei nuclei in North China, as well as the Mairi gneiss complex in the São Francisco Craton[49]; (ii) similar c. 2.7 Ga basaltic volcanism, including komatiite, is preserved within the North China Craton[50] and elsewhere in Sclavia supercraton[51,52]; (iii) North China and other cratons of the Sclavia supercraton share similar cratonization ages during the late Neoarchaean (2.6–2.5 Ga; refs. 51,53); and (iv) both North China and the Sclavia supercraton lack early Paleoproterozoic glaciogenic sequences, which are characteristic of the Superior-like cratons[45] (whereas deposits of the Hutuo Group of North China are of ambiguous glacial origin and likely a distinctly younger age than the Huronian Group of Superior[54]). Together, our new data suggest that North China belonged to the Sclavia supercraton, and affirm previous hypotheses that the supercontinent-like cycle of continental assembly has operated on Earth since at least the late Archaean.

Our new data from the North China Craton can further shed light on a key question within the geosciences: when did the transition between a stagnant lid and mobile lid (plate tectonic) geodynamic regime occur on Earth? Zircon oxygen isotope analysis has been proposed to track subduction zone processes[55]. In this study, we compiled a dataset that includes new zircon oxygen isotopes and data from various terranes within North China, as well as other cratons worldwide. Globally, >2.7 Ga zircons primarily display mantle-like or slightly elevated $\delta^{18}O$ values, whereas those with ages <2.7 Ga show an increasing trend in $\delta^{18}O$ values (Fig. 2f), as similarly demonstrated in previous analyses[41]. In North China specifically, zircon $\delta^{18}O$ values only begin to show significant elevation by the end of the Neoarchaean (c. 2.5 Ga), while 3.6–2.6 Ga zircon exhibit mantle-like or only slightly elevated $\delta^{18}O$ values (Fig. 2f). Significant

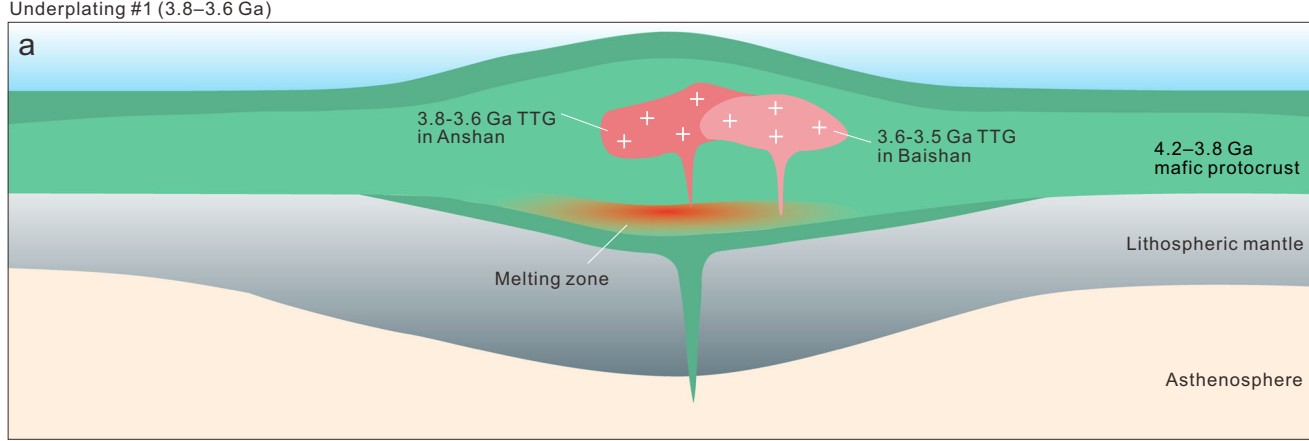

Underplating #1 (3.8–3.6 Ga)

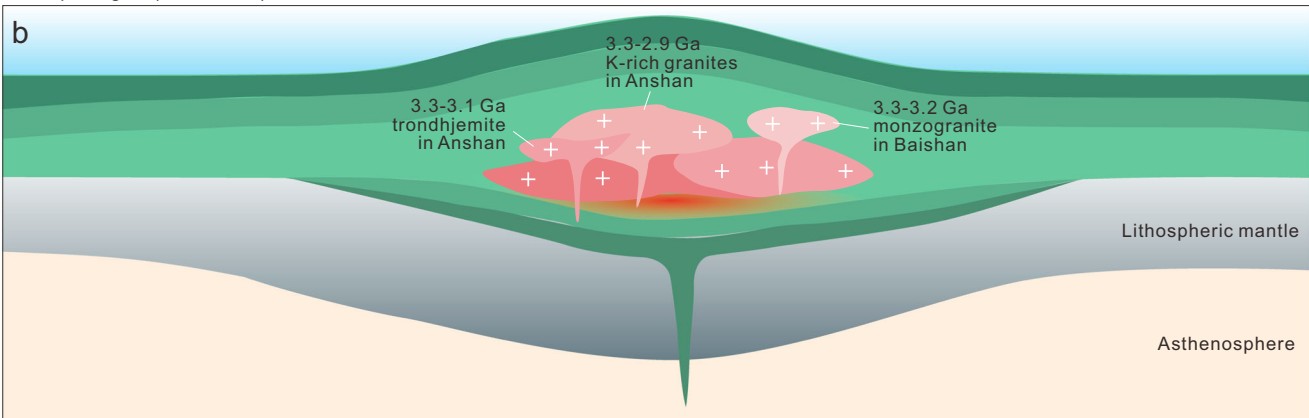

Underplating #2 (3.3–2.9 Ga)

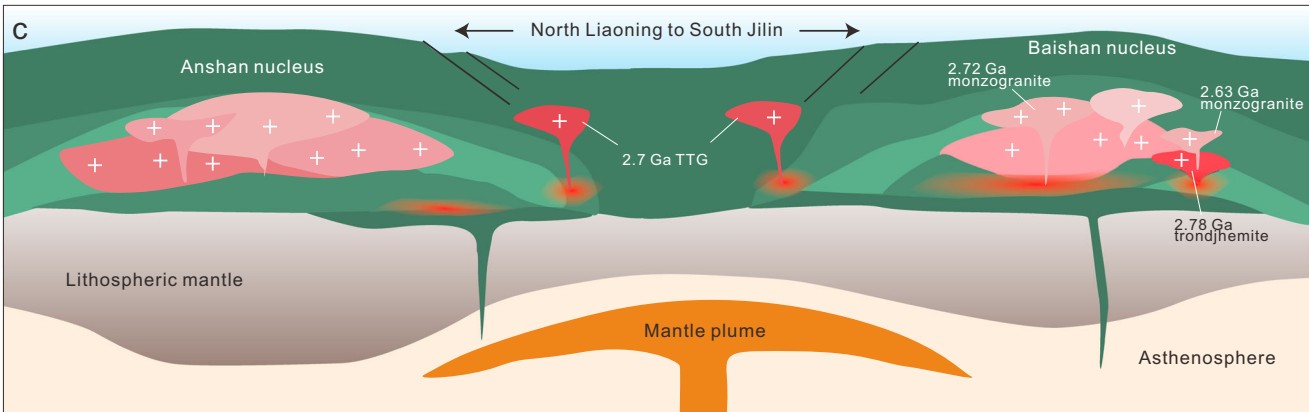

Underplating #3 (2.8–2.6 Ga)

**Fig. 5 | Tectono-magmatic model of the Eoarchaean to early Neoarchaean evolution of the North China Craton. a** Mafic magma underplating during 3.8–3.6 Ga resulted in partial melting of a pre-existing 4.2–3.8 Ga mafic protocrust to generate 3.8–3.6 Ga TTG and form the earliest coherent continental nucleus in the North China Craton. **b** Another mafic magma underplating event during 3.3–2.9 Ga led to the reworking of previously formed 3.8–3.6 Ga felsic crust. As a result, abundant 3.3–2.9 Ga potassium-rich granites formed in the Anshan and Baishanhu nuclei. **c** The 2.8–2.6 Ga magma underplating event led to the reworking of 3.8–3.6 Ga ancient felsic crust and 3.2–2.9 Ga juvenile mafic crust, which resulted in the formation of 2.72 Ga potassium-rich granite in the Baishanhu nucleus and c. 2.7 Ga TTG in the North Liaoning to South Jilin granite-greenstone belt, respectively. The green layers in the models represent mafic crusts formed during different stages.

elevation of $\delta^{18}O$ values during the Neoarchaean is consistent with the observation from triple-oxygen-isotopes recorded in Archaean shales[56] that reflect substantial emergence of subaerial landmass, allowing for subaerial weathering. The appearance of more exposed continent during the Neoarchaean is contemporaneous with the assembly of the Kenorland supercontinent[42] or multiple supercratons[43] (i.e., Superia, Sclavia, and Vaalbara). Thus, the diachronous increase of $\delta^{18}O$ values in different cratons at different times in the Neoarchaean suggests the asynchronous emergence of large-scale subaerial land.

There is broad consensus that plate tectonics had become established on Earth by around 3 Ga (refs. 1,57,58), even if localised subduction began in some cratons at an earlier time[59–61]. The $\delta^{18}O$ values of the c. 2.5 Ga igneous rocks from North China show notably higher values than those in older zircon (Fig. 2f), which suggest that considerable volumes of supracrustal materials have been

incorporated into magma source regions. This process can be achieved through several geodynamic processes (e.g., subduction, sagduction, or thrust stacking), of which, subduction is considered the most effective way. Several independent lines of evidence confirm that subduction initiated locally within the North China Craton by the end of the Neoarchaean, including evidence of paired metamorphism in Dengfeng[62], Alpine-style subhorizontal arc-affinity nappe structures in central North China[63,64], and the widespread presence of subduction-related potassic granites and sanukitoids[53]—all occurring at c. 2.5 Ga. Therefore, the initiation of subduction in North China might have occurred notably later than in many other cratons. Our new data thus add to a growing set of observations supporting the onset of sub-duction being asynchronous from a global perspective[59,65]. This later tectono-magmatic age is interestingly also when the global $\delta^{18}O$ database on the whole exhibits a positive step-change increase towards more supracrustal reworking taken to indicate a geodynamic shift into the "supercontinent state", whereafter the three large supercontinents are known to have formed[41]. This relationship underscores how North China Craton represents one of the last places in a globally highly asynchronous process to have experienced the onset of subduction.

## Methods

### Zircon U–Pb dating

Zircon U–Pb analysis of samples 21LJ38-1, 21LJ39-1, 21LJ35-1, 21LJ06-3, 21LJ18-1, and 21LJ19-1 was conducted using a PlasmaQuant MS series ICP–MS with a NWR193 laser-ablation microprobe at Yanduzhongshi Geological Analysis Laboratories Ltd, Beijing, China. The diameter of laser beam was set at 30 μm. The analysis of samples 22BS20-3, 22BS15-1, 22BS19-1, 22BS23-1, 22BS27-1, 22BS28-1, 22BS29-1, 22BS32-1, 22BS18-4, 23HX22-1, 23HX23-1, 23HX28-1, and 23HX29-1 was carried out using an Agilent 7500c quadrupole ICP–MS and a 193-nm ArF Excimer laser at the Key Laboratory of Mineral Resources Evaluation in Northeast Asia at Jilin University, Changchun, China. The analytical spot size was set at 32 μm, with a laser energy density of 10 J/cm$^2$ and a repetition frequency of 8 Hz. Zircon U–Pb dating of Sample Z2010-3 was per-formed using an Agilent 7900 ICP–MS with an ATL (ATLEX 300) excimer laser at Nanjing Hongchuang Exploration Technology Service Co., Ltd., Nanjing, China. Zircon standards 91500 and Plešovice were utilised as primary and secondary reference materials in all three above laboratories, respectively. In addition, Samples 21LJ39-1, 22BS20-3, and 22BS15-1 were also analyzed using the Sensitive High Resolution Ion MicroProbes (SHRIMP) II instrument at the Beijing SHRIMP Center of the Chinese Academy of Geological Sciences (GAGS), Beijing, China. During the analysis, the intensity of the primary O$^{2-}$ ion beam was set to 3–5 nA, with beam spot sizes of 20 μm. Standard zircon TEMORA was utilised to correct the U, Th, and Pb contents as well as the ages of zircon. The detailed analytical procedures for the zircon U–Pb isotopes are presented in Supplementary Methods.

### Zircon Lu–Hf isotope analysis

In situ Lu–Hf isotope ratio analysis was conducted using a Neptune Plus MC–ICP–MS in conjunction with a Geolas HD excimer ArF laser ablation system at the Wuhan Sample Solution Analytical Technology Co., Ltd, Wuhan, China. All data were collected on zircon in single spot ablation mode at a spot size of 44 μm. The energy density of laser ablation used in this study was approximately 10 J cm$^{-2}$. Each mea-surement consisted of a 20-s acquisition of the background signal followed by a 50-s acquisition of the ablation signal. Plešovice was utilised for external standard calibration to optimise the analytic results. 91500 and GJ-1 served as secondary standards to monitor the quality of data correction. The $\varepsilon Hf_{(t)}$ values are calculated using $^{176}Lu$ decay constant ($\lambda^{176}Lu$) of $1.865 \times 10^{-11}$ (ref. 66) and the chondrite parameters used are $^{176}Hf/^{177}Hf = 0.282772$ and $^{176}Lu/^{177}Hf = 0.0332$ (ref. 67). The zircon Hf two-stage depleted mantle model ages ($T_{DM}^2$)

were calculated using ratios of $^{176}Hf/^{177}Hf = 0.283251$ and $^{176}Lu/^{177}Hf = 0.0384$ for present depleted mantle suggested by ref. 68, and $^{176}Lu/^{177}Hf$ value of 0.022. The detailed analytical procedures for the zircon Lu–Hf isotopes are described in Supplementary Methods.

### Major and trace element analyses

Whole-rock major and trace element analyses were conducted at Wuhan Sample Solution Analytical Technology Co., Ltd, China. The Zsx Primus II wavelength dispersive X-ray fluorescence spectrometer was used to analyze the major elements. The standard curves were derived using the national standard materials GBW07103, GBW07105, GBW07111, and GBW071112. The relative standard deviation was less than 2%. An Agilent 7700e ICP-MS equipment was used to analyze the trace elements. The standard materials GSR-3, RGM-2, BHVO-2, and JA-2 were used for quality control. The detailed analytical procedures for the whole-rock geochemistry are described in Supplementary Methods.

### Zircon O isotope analysis

The analysis of zircon oxygen isotope was conducted using the SHRIMP II equipment at the Beijing SHRIMP Center, CAGS, China. The Cs$^+$ primary ion beam's intensity was approximately 3 nA, which resulted in secondary $^{16}O^{1-}$ count rates exceeding $10^9$ cps. The dia-meter of the spot analyzed was 20 μm. The standard zircon TEMORA was used as reference material for calibrating instrumental mass fractionation. The standard was analyzed either two or three times at the beginning of each analytical session, and then after every third analysis of the unknown samples. The detailed analytical procedures for the zircon O isotopes are presented in Supplementary Methods.

### Compilation of zircon Hf–O data

Zircon Hf isotopic data have been compiled from various sources for the continental nuclei of the NCC and the Neoarchaean northern Liaoning to southern Jilin granite-greenstone belt. Additionally, Eoarchaean-Mesoarchaean TTG samples from other cratons have also been included in this compilation. These cratons include: (1) India: Bastar Craton, Bundelkhand Craton, Coorg Block, Dharwar Craton, and Singhbum Craton; (2) South Africa: Kaapvaal Craton; (3) North America: North Atlantic Craton, Slave Craton, Superior Craton, and Wyoming Craton; (4) Australia: Pilbara Craton and Yilgarn Craton; (5) China: Yangtze Craton and Tarim Craton; (6) Other locations: East Antarctica and Sao Francisco Craton (South America). The compiled zircon Hf isotopic data and references are presented in Supplementary Data 4. This compilation includes only the data of magmatic zircons with U–Pb age discordances <10%. Generally, crustal contamination or magma mixing could induce heterogeneous Hf isotope signatures. Thus, this study applies probability of fit ($p$) values of each sample to filter out those zircons with heterogeneous Hf compositions. The $p$ values of each sample were calculated using Origin 2023. Only samples with $p \geq 0.05$ are considered statistically homogeneous, while those with $p < 0.05$ are deemed heterogeneous and thus excluded from interpretation. The in situ zircon O isotopic data have been compiled from various terranes of the NCC and other ancient cratons/terranes worldwide. These cratons/terranes include the Slave craton, Barber-ton, Clearwater Block, Congo Craton, Coorg Block, Dharwar craton, East Antarctica, North Atlantic craton, Pilbara craton, Scandinavia, Superior craton, Tarim craton, Yangtz craton, and Yilgarn craton. The compiled zircon $\delta^{18}O$ values and references for these data are provided in Supplementary Data 5. Only analyses for magmatic zircons or cores with U–Pb age discordances <10% were included.

## Data availability

The authors declare that all data supporting the findings of this study are available online (https://doi.org/10.6084/m9.figshare.26139169) and included in Supplementary Information/Data files.

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

## Acknowledgements

This work was funded by the National Natural Science Foundation of China (NSFC, 42025204 to J.Z., 42172212 to J.L., 42272224 to Z.H.L, and U2244211).

## Author contributions

J.L. and R.M.P. designed the project. J.L., Z.S.L., C.Q.C., and H.X.Z. participated in the fieldwork and analysis. J.L. wrote the origin draft. R.M.P., R.N.M., J.Z., and Z.H.L. edited and commented on the draft.

## Competing interests

The authors declare no competing interests.
