## [Peer Review File · Nature Communications]

REVIEWER COMMENTS

Reviewer #1 (Remarks to the Author):

Please the file titled 'Review_document.docx' attached.

Reviewer #2 (Remarks to the Author):

This manuscript presents new zircon U-Pb age, Lu-Hf isotope and Oxygen isotope data for the newly recognized 3.57 Ga Tonalite, together with other younger rocks, in the North China Craton. It represents the finding of a new continental nucleus in the North China Craton. Based on the data, the authors addressed three problems: the widespread of Hadean to early Eoarchean proto-crust, the geodynamic process for the Paleoproterozoic to Mesoproterozoic magmatism and the initiation of plate tectonics. The data is interesting and the manuscript is well organized. This research is generally of broad interests for researchers who are interested in the early earth. I suggest the authors add some new issues in the revision. First, the readers would like to know if the Eoarchean to Paleoproterozoic rocks in the Jiapigou–Baishan Lake area of this study, and those reported in previous studies, mainly in the Anshan-Benxi area and the eastern Hebei area belong to the same coherent continental nucleus or three individual nuclei. The other issue is about the geodynamic process of the 3.57-2.72 Ga magmatism. The authors proposed the 3.57-2.72 Ga magmatism, including tonalite, monzogranite and trondhjemite, were formed by repeated plume underplating in a stagnant lid background, mainly based on their Hf isotopes. This seems acceptable but not enough. The readers would wonder what controls the major and trace element geochemical characteristics (which is missing in the presented data) of the 3.57 Ga tonalite, 3.23 Ga monzogranite, 2.78 Ga trondhjemite, 2.72 Ga monzogranite if they are all derived from the same proto-crust. What are the differences in their melting conditions? What are the implications of such differences? And I disagree with the authors on their discussion on the onset of plate tectonics. The authors proposed asynchronous onset of plate tectonics in different cratons based on asynchronous increase of Hf isotope and O isotopes. In my opinion, the asynchronous onset of plate tectonics is a fake proposition because plate tectonics is defined to be a global tectonics characterized by the creation and maintenance of a global network of narrow plate boundaries. The increase of Hf isotope ratios may signify the contribution of depleted mantle related to plate subduction. The increase of O isotope ratios suggests the emergence of large-scale subaerial landmass for low-temperature weathering.

Below are detailed comments and suggestions.

Line 121: The title is not consistent with the contents of this section. I can only see the crustal architecture of northeastern NCC from this paragraph. I can see nothing about the structure of lithospheric mantle in the whole manuscript.

Lines 127-128: You need to provide the ages of these so-called “old crust”. Only unradiogenic $\epsilon\text{Hf}(t)$ values do not support they are very old.

The effective numbers: For example, the uncertainty of $\epsilon\text{Hf}(t)$ values is about 0.4-0.5. Therefore, the

significant digit after the decimal point of $\epsilon\text{Hf}(t)$ should be 1. And the significant digit of $^{176}\text{Hf}/^{177}\text{Hf}$ and error is traditionally 6.

Fig. 3a: The design of this figure is not logical. The hot and cold color is designed to represent negative and positive $\epsilon\text{Hf}(t)$ values. However, the same $\epsilon\text{Hf}(t)$ values at different ages represent different mantle extraction age. If they are labelled as the same color, the authors would get the wrong information that the mantle extraction ages are the same. I strongly suggest the authors provide zircon U-Pb age map and TDM2 map in Fig. 3.

Lines 146-148: It is impossible for the existence of a “global-scale” Hadean to early Eoarchean proto-crust. This means that almost all continental crust were formed in the very early stage of the Earth history. If so, most of the continental crust should have modal age of Hadean to early Eoarchean, which was not observed in zircon Hf model age data base (e.g., Dhuime et al., 2012).

Dhuime, B., Hawkesworth, C.J., Cawood, P.A., Storey, C.D., 2012. A change in the geodynamics of continental growth 3 billion years ago. *Science* 335, 1334-1336.

Lines 152-154: The statement is not consistent with Fig. 2b. Not all the >2.6 Ga granitoids within the ABN exhibit a simple linear evolutionary ϵHf value trend. The 3.1-3.0Ga rocks in the ABN show obvious increase in ϵHf values.

Lines 160-161: I cannot see the crustal architecture characterized by “younger and juvenile granite-greenstone belts surrounding ancient, long-lived, and reworked continental nuclei” from Fig. 3.

Lines 195-206: I personally agree that the NCC may belong to the Sclavia supercraton. But I cannot agree with the logics of the first evidence. The identification of Hadean-Eoarchean rocks and detrital zircons in these cratons, do not mean that they are in the same supercraton in the Neoarchean. The Hadean detrital zircons might be transported to other cratons at any time if they were in the same supercontinent.

Lines 213-215: Subduction is not a pre-requirement for the increase of $\delta^{18}\text{O}$ values. Other geodynamic processes for recycling of supracrustal materials (e.g., sagduction) are also capable to incorporate supracrustal materials into magmatic sources. The key point is the emergence of subaerial landmass for low-temperature weathering. Therefore, the synchronous increase of $\delta^{18}\text{O}$ values in different cratons suggest the asynchronous emergence of large-scale subaerial land rather than asynchronous initiation of subduction.

Bindeman, I.N., Zakharov, D.O., Palandri, J., Greber, N.D., Dauphas, N., Retallack, G.J., Hofmann, A., Lackey, J.S., Bekker, A., 2018. Rapid emergence of subaerial landmasses and onset of a modern hydrologic cycle 2.5 billion years ago. *Nature* 557, 545-548.

Line 225: onset of plate tectonics or onset of subduction? Plate tectonics is defined to be a global tectonics characterized by the creation and maintenance of a global network of narrow plate boundaries. According to its definition, if plate tectonics begun, it is globally. Therefore, you can say asynchronous onset of subduction but cannot say asynchronous onset of plate tectonics.

GPS coordinates should be provided in the supplementary tables.

The age of 22BS20-3: From the field photo of Fig. S3b and d, I cannot see any difference between the 3.57 Ga tonalite and the 3.2 Ga monzogranite. There are three groups of ages in Fig. S6a and b, which group really record the crystallization age? To support your interpretation that the 3.57 Ga group is the crystallization age, you need to provide more zircon CL images and describe their Th/U ratios.

Lu-Hf isotope and Oxygen isotope results for the metamorphic domains should not be plotted on the figures. If the zircon metamorphic domain is of recrystallization origin, its $^{176}\text{Hf}/^{177}\text{Hf}$ isotope ratios should be similar with its igneous predecessor whereas its U-Pb age is reset. If the Hf isotope is plotted on the $e\text{Hf}(t)$ -age diagram, it will mislead the readers that the magmatism of this age have such $e\text{Hf}(t)$ values.

Review of Liu et al., - Nature Communications (January '24)

This manuscript presents new zircon U-Pb, Hf and O isotopic data from the Jiapigou–Baishan Lake nucleus (JBLN) of the North China Craton (NCC). Hf isotopic compositions of zircon from Paleoproterozoic tonalitic rocks and zircon xenocrysts in younger potassic granites indicate repeated reworking of 3.8–4.2 Gyr old mafic source rock over a period of 1 billion years. This data is inferred to support the existence of a Hadean to early Eoarchean heritage for the NCC and the long history of crustal reworking is inferred to reflect repeated remelting in an intraplate/stagnant lid environment. Overall, I found the manuscript well written and easy to understand. However, there are a few issues that need to be addressed.

The U-Pb and Hf isotope data for the standard materials are not reported in the supplementary dataset so it is not possible to verify how robust the data are. Please report Hf isotope data for all standard material and unknowns, preferably in a format similar to Table 3 in (Fisher et al., 2014), that includes the stable Hf isotope ratios (e.g. $^{178}\text{Hf}/^{177}\text{Hf}$ and/or $^{180}\text{Hf}/^{177}\text{Hf}$), so that it is easy to assess the quality of the data. There appears to be a plotting / normalisation error with the Hf isotope data. On the diagram of $^{176}\text{Hf}/^{177}\text{Hf}$ versus Age shown in Fig. S1 (supplementary material), the 3.5–3.6 Ga rocks cluster above the CHUR model evolution line, whereas in Fig 2b of the main text the same rocks plot below the CHUR model evolution line (please see the comparison in Figure below). Whilst this doesn't necessarily influence the inferred tectonic history, it needs to be investigated because it could have a profound influence on the apparent model ages and need for a Hadean source, which is a significant part of the novelty of this study. Based on Fig S1, it doesn't appear as though there is any need for a Hadean source.

Further to this point, Hf isotope model ages are fraught with uncertainty and very sensitive to the assumed mantle depletion history. Although there is some justification for the chosen mantle depletion history in the supplementary information. I think there needs to be a more nuanced discussion around the calculation of Hf isotope model ages (after the issue above has been addressed) and the inferred mantle depletion history in the main text.

While I tend to agree with the authors view that the development of plate tectonics was highly asynchronous globally, I think that if one takes this stance, it is also reasonable to think that the history of mantle depletion could have progressed in a similar fashion. My sense is that modelling mantle depletion as a single global event at 4.5 Ga is a bit simplistic, and doesn't consider more recent work attempting to address this issue through Hf-Nd isotope systematics in ancient crustal rocks (Kemp et al., 2023; Petersson et al., 2019). For example, in East Pilbara Terrane of the Pilbara Craton — which is arguably the archetypal example of intraplate continental growth processes (Van Kranendonk et al., 2007) — it appears as though there is a relatively long history of chondritic magmatism throughout the Paleoproterozoic, with evidence of only moderate mantle depletion starting around 3.8 Ga (Kemp et al., 2023). Indeed, in Figure 2b it appears like the preferred crustal evolution line ($^{176}\text{Hf}/^{177}\text{Hf} = 0.022$) for the most evolved zircon Hf compositions intersects the chondrite evolution curve at a slightly younger date (around 3.9–4.0 Ga) and that a mantle depletion history starting around this time still provides a reasonable fit for the most juvenile compositions observed in the NCC (see modified version of Figure 2b below). Such a depletion history will shift the model ages to younger values. Provided the plotting / normalisation issue isn't to blame it might not affect the need for a Hadean source, but this needs to be properly evaluated. I think having some discussion around this is warranted and will ultimately make for a more robust and hopefully compelling discussion.

Another point to consider is that the data point yielding the inferred Hadean age is not from the 3.5–3.6 tonalites (partial melts of mafic crust), but rather from the c. 3.2 Ga

monzogranites. Again, this may be influenced by the plotting/normalisation issue, but it is also worth considering if it is justified to model Hf isotope evolution using a single stage model with mafic composition, when the petrology of the rocks requires at least two stage evolution, involving reworking of more evolved (felsic) material?

Finally, it strikes me that ^{142}Nd isotope data on some of these rocks could provide a critical test for the hypothesis presented in this manuscript. Are any such data from the JBLN or one of the other cratonic nuclei in the NCC available? If so, it could be useful to cite as additional evidence for a Hadean source if there is some.

Additional minor comments in the attached pdfs.

References

- Fisher, C. M., Vervoort, J. D., & Hanchar, J. M. (2014). Guidelines for reporting zircon Hf isotopic data by LA-MC-ICPMS and potential pitfalls in the interpretation of these data. *Chemical Geology*, 363, 125-133.
- Kemp, A. I., Vervoort, J. D., Petersson, A., Smithies, R. H., & Lu, Y. (2023). A linked evolution for granite-greenstone terranes of the Pilbara Craton from Nd and Hf isotopes, with implications for Archean continental growth. *Earth and Planetary Science Letters*, 601, 117895.
- Petersson, A., Kemp, A. I., Hickman, A. H., Whitehouse, M. J., Martin, L., & Gray, C. M. (2019). A new 3.59 Ga magmatic suite and a chondritic source to the east Pilbara Craton. *Chemical Geology*, 511, 51-70.
- Van Kranendonk, M. J., Smithies, R. H., Hickman, A. H., & Champion, D. C. (2007). Paleoproterozoic development of a continental nucleus: the East Pilbara terrane of the Pilbara craton, Western Australia. *Developments in Precambrian Geology*, 15, 307-337.

Comments in black

Responses in blue

Revisions in red

Line numbers refer to the “clean” version of the manuscript *without* tracked changes.

Reviewers’ comments:

Reviewer #1 (Remarks to the Author):

This manuscript presents new zircon U-Pb, Hf and O isotopic data from the Jiapigou–Baishan Lake nucleus (JBLN) of the North China Craton (NCC). Hf isotopic compositions of zircon from Paleoproterozoic tonalitic rocks and zircon xenocrysts in younger potassic granites indicate repeated reworking of 3.8-4.2 Gyr old mafic source rock over a period of 1 billion years. This data is inferred to support the existence of a Hadean to early Eoarchean heritage for the NCC and the long history of crustal reworking is inferred to reflect repeated remelting in an intraplate/stagnant lid environment. Overall, I found the manuscript well written and easy to understand. However, there are a few issues that need to be addressed.

Response: We sincerely thank the reviewer for this comprehensive assessment.

Comment 1:

The U-Pb and Hf isotope data for the standard materials are not reported in the supplementary dataset so it is not possible to verify how robust the data are. Please report Hf isotope data for all standard material and unknowns, preferably in a format similar to Table 3 in (Fisher et al., 2014), that includes the stable Hf isotope ratios (e.g. $^{178}\text{Hf}/^{177}\text{Hf}$ and/or $^{180}\text{Hf}/^{177}\text{Hf}$), so that it is easy to assess the quality of the data.

Revision: We appreciate the reviewer’s attention to the data of standard materials. We have added U–Pb and Hf isotope data for all standard materials at the end of Supplementary Table S1-3, respectively. We also provide weighted average diagrams of the U–Pb and Hf isotope data of standard materials for easily assessing the quality of the data. In addition, we added the stable Hf isotope ratios ($^{178}\text{Hf}/^{177}\text{Hf}$) for all standard materials and unknowns (Supplementary Table S3), and reformatted the Hf isotope data according to Fisher et al. (2014) as the reviewer suggested. LA–ICP–MS U–Pb data of standard materials are provided at the end of Supplementary Table S1. See the following image below as a demonstration.

Ga rocks cluster above the CHUR model evolution line, whereas in Fig 2b of the main text the same rocks plot below the CHUR model evolution line (please see the comparison in Figure below). Whilst this doesn't necessary influence the inferred tectonic history, it needs to be investigated because it could have a profound influence on the apparent model ages and need for a Hadean source, which is a significant part of the novelty of this study. Based on Fig S1, it doesn't appear as though there is any need for a Hadean source.

Revision: We greatly appreciate the reviewer for pointing out this plotting error in Fig. S1 (previous). We regret that we incorrectly used the present (i.e., dated) $^{176}\text{Hf}/^{177}\text{Hf}$ ratios rather the initial (i.e., calculated) ratios that should be used in the diagram of $^{176}\text{Hf}/^{177}\text{Hf}$ versus age. According to the new diagram of $^{176}\text{Hf}/^{177}\text{Hf}$ versus age (new Fig. 2e), the 3.5–3.6 Ga rocks of this study cluster below the CHUR model evolution line, which is consistent with the $\epsilon_{\text{Hf}(t)}$ versus age diagram (new Fig. 2d). Meanwhile, the depleted mantle model ages of these 3.6–2.7 Ga zircons indeed suggest a Hadean to early Eoarchaean (4.2–3.8 Ga) source. In addition, several Eoarchaean zircons from the Anshan nucleus and Eastern Hebei nucleus show even lower $\epsilon_{\text{Hf}(t)}$ values than the 3.6 Ga zircons of this study. This strongly suggests a Hadean protocrust (considered the mafic precursors derived from chondritic mantle) once existed in the NCC. Also, in case some additional geological signals can be detected, we plot all analyses rather than the mean values of each sample in new diagrams of $^{176}\text{Hf}/^{177}\text{Hf}$ vs. Age and $\epsilon_{\text{Hf}(t)}$ vs. Age. (Those zircon with obvious Pb loss were excluded.) In addition, those zircon with heterogeneous Hf compositions (probability of fit (p) values < 0.05), which could be due to contamination, magma mixing, later metamorphism and/or alteration, were not considered in the interpretation. These heterogeneous zircons are plotted in the diagrams, but are displayed in light gray. Please see the comparison in the figures below.

Comment 3:

Further to this point, Hf isotope models ages are fraught with uncertainty and very sensitive to the assumed mantle depletion history. Although there is some justification for the chosen mantle depletion history in the supplementary information. I think there needs to be a more nuance discussion around the calculation of Hf isotope models ages (after the issue above has been addressed) and the inferred mantle depletion history in the main text. While I tend to agree with the authors view that the development of plate

tectonics was highly asynchronous globally, I think that if one takes this stance, it is also reasonable to think that the history of mantle depletion could have progressed in a similar fashion. My sense is that modelling mantle depletion as a single global event at 4.5 Ga is a bit simplistic, and doesn't consider more recent work attempting to address this issue through Hf-Nd isotope systematics in ancient crustal rocks (Kemp et al., 2023; Petersson et al., 2019). For example, in East Pilbara Terrane of the Pilbara Craton — which is arguably the archetypal example of intraplate continental growth processes (Van Kranendonk et al., 2007) — it appears as though there is a relatively long history of chondritic magmatism throughout the Paleoproterozoic, with evidence of only moderate mantle depletion starting around 3.8 Ga (Kemp et al., 2023). Indeed, in Figure 2b it appears like the preferred crustal evolution line ($^{176}\text{Lu}/^{177}\text{Hf} = 0.022$) for the most evolved zircon Hf compositions intersects the chondrite evolution curve at a slightly younger date (around 3.9–4.0 Ga) and that a mantle depletion history starting around this time still provides a reasonable fit for the most juvenile compositions observed in the NCC (see modified version of Figure 2b below). Such a depletion history will shift the model ages to younger values. Provided the plotting/normalisation issue isn't to blame it might not affect the need for a Hadean source, but this needs to be properly evaluated. I think having some discussion around this is warranted and will ultimately make for a more robust and hopefully compelling discussion.

Response: We greatly appreciate the reviewer's concern about the Hf isotope model age. We agree with the reviewer that more discussion about the mantle depletion history will make the interpretation more robust and convincing. Firstly, we moved the related discussion previously in the supplementary material into the main text now. Then, we compiled a new zircon Hf isotopic dataset of early Archaean TTG from cratons worldwide. This new compilation can give us more information about the crustal growth history of early Earth and provide some hints about the crustal residence time of the mafic precursors of these early Archaean TTGs. This compilation suggests long crustal residence times for these ancient TTGs, which can, as revealed by Liou et al. (2022)¹, potentially lead to significant underestimation of zircon $\epsilon\text{Hf}(t)$ values and the degree of initial mantle depletion. Additionally, our new compilation of Archaean zircon Hf isotopes from the North China Craton suggests that a considerable proportion of Eoarchaean zircon have depleted Hf isotopes. Combined with the ^{142}Nd – ^{143}Nd isotopes, such zircon Hf signatures demonstrate the existence of depleted mantle in the North China Craton during the Eoarchaean. Thus, collectively, the evidence available globally at this time appears to suggest that both depleted and chondritic mantle existed in different domains on Earth during the early Archaean. The related detailed text revisions are as follows.

Revision: Lines 93-124 “however, we note that the Hf T_{DM}^2 age is affected by the assumed mantle depletion history and the $^{176}\text{Lu}/^{177}\text{Hf}$ ratio of the crust. Currently, the community's understanding of the evolution of the Earth's depleted mantle through time is uncertain. Several studies of Eoarchaean TTGs (e.g. the Itsaq gneiss, North Atlantic Craton², and the Aktash gneiss, Tarim Craton³) have indicated that their mafic precursors were derived from a near-chondritic mantle. Further, Hf–Nd isotopes of Palaeoarchaean (c. 3.6 Ga) mafic to ultramafic rocks from the Pilbara Craton also indicate the existence of a chondritic to near-chondritic mantle at that time^{4, 5}. Thus, these authors interpret that

any global depletion of the mantle had not begun during the Eoarchaeon or Hadean, such that a depleted mantle signature began to progressively develop since c. 3.9 Ga (ref. ⁵), although see ref. ¹ for the potential caveats involved when zircon is used to estimate mantle depletion.

To directly address this issue, we compiled a global dataset of igneous zircon Hf isotopes of early Archaean TTGs from all major cratons on Earth (**Table S4**). The compilation reveals that most zircons display very low sub-chondritic $\epsilon_{\text{Hf}(t)}$ values (**Fig. 2c**), indicating that the mafic precursors of these early Archaean TTGs likely had a long crustal residence time, potentially leading to significant underestimation of zircon $\epsilon_{\text{Hf}(t)}$ value and the degree of mantle depletion¹. Several Palaeoarchaeon mafic–komatiitic (e.g., Barberton Greenstone Belt⁶ and Western Pilbara Craton⁷) and TTG rocks (e.g., Dharwar Craton⁸ and Singhbhum Craton⁹) that display depleted mantle signatures also argue for the existence of a depleted mantle in the early Archaean. After excluding analyses that show clear Pb loss and samples with heterogeneous Hf isotopic compositions, many Eoarchaeon zircons from North China still record highly depleted Hf isotope signatures (**Fig. 2d**). Measured ¹⁴²Nd–¹⁴³Nd isotopes of c. 3.8–3.0 Ga rocks from the Anshan Complex suggest multiple mantle–crust differentiation events between 4.3 Ga and 3.8 Ga, and indicate the existence of a depleted upper mantle during this period¹⁰. Thus, we interpret that the early Archaean mantle was heterogeneous with some domains (e.g., North China and Dharwar Cratons) that exhibited depleted Hf isotopes, whereas other domains (e.g., North Atlantic and Tarim Cratons) exhibited chondritic compositions. Therefore, the T_{DM^2} age calculation performed in this study assumes that early Archaean mafic protocrust within North China formed from a depleted mantle. Additionally, a plot of zircon ages against their initial ¹⁷⁶Hf/¹⁷⁷Hf ratios demonstrates that the 3.6–2.7 Ga zircon from the Baishan nucleus, the 3.8–2.7 Ga zircon from the Anshan nucleus, and the Eoarchaeon–Palaeoarchaeon detrital zircon of the Caozhuang sequence from the Eastern Hebei nucleus all fall along a crustal evolution line with a ¹⁷⁶Lu/¹⁷⁷Hf ratio of 0.022 (**Fig. 2e**). This trend is consistent with the ¹⁷⁶Lu/¹⁷⁷Hf ratio observed in Archaean mafic crust¹¹. Thus, we consider it reasonable to use a ¹⁷⁶Lu/¹⁷⁷Hf ratio of 0.022 for calculating the T_{DM^2} ages of the Archaean zircon from North China.”

Comment 4: Another point to consider is that the data point yielding the inferred Hadean age is not from the 3.5–3.6 tonalites (partial melts of mafic crust), but rather from the c. 3.2 Ga monzogranites. Again, this may be influenced by the plotting/normalisation issue, but it is also worth considering if it is justified to model Hf isotope evolution using a single stage model with mafic composition, when the petrology of the rocks requires at least two stage evolution, involving reworking of more evolved (felsic) material?

Response: We appreciate the reviewer’s concern about the inferred Hadean age. Firstly, we resolved the plotting issue mentioned in “Comment 2”. And we have a comprehensive discussion on the calculation of depleted mantle model ages. It is reasonable to regard two-stage depleted mantle ages as representing the mantle extraction ages of the mafic precursors of these granitoids. Almost all the 3.6–2.7 Ga zircons yield 4.2–3.8 Ga T_{DM^2} ages, not only the 3.2 Ga monzogranites. These 3.6–2.7 Ga zircons fall between the 4.2 Ga and 3.8 Ga crustal evolution lines on the $\epsilon_{\text{Hf}(t)}$ vs. Age diagram (Fig. 2d; refer to following figure). We agree with the reviewer that single stage model age is suitable for

mafic rocks, whereas a two-stage model age is suitable for felsic rocks. So, following this principle to make the contour map of T_{DM} age (Fig. 4b; see below), we chose a single-stage model age for meta-mafic rocks and a two-stage model age for TTGs and other granitoids.

Fig. 2d

Fig. 4b

Comment 5: Finally, it strikes me that ^{142}Nd isotope data on some of these rocks could provide a critical test for the hypothesis presented in this manuscript. Are any such data from the JBLN or one of the other cratonic nuclei in the NCC available? If so, it could be useful to cite as additional evidence for a Hadean source if there is some.

Response: We appreciate the reviewer for this useful suggestion. Currently, only one paper (Li et al., 2014) reported ^{142}Nd and ^{143}Nd isotope data for the 3.8–3.0 Ga Anshan Complex in the Anshan–Benxi nucleus. Their research also suggests a Hadean evolution history for the NCC. We added this evidence in the main text.

*Li, C.-F. et al. Differentiation of the early silicate Earth as recorded by ^{142}Nd - ^{143}Nd in 3.8–3.0 Ga rocks from the Anshan Complex, North China Craton. *Precambrian Research* **301**, 86-101 (2017).*

Revision: Lines 215-217 “ ^{142}Nd and ^{143}Nd isotopes for the 3.8–3.0 Ga Anshan Complex suggest 4.5–4.4 Ga model ages for the precursor of the oldest components and multiple mantle-crust differentiation events from 4.3 to 3.8 Ga (ref. ¹⁰).”

Additional minor comments in the attached pdfs:

1# Line 25-26 “*primarily owing to the paucity of well-preserved crustal remnants from this eon.*”: Reads as though Archean rocks in general are not well preserved. If you are referring to Eo-to-Paleoarchean 'protocrust' (crustal nuclei) I think it would be to be a bit more specific.

Response: We appreciate the reviewer for pointing out this ambiguous expression. We agree with you that it is better to be more specific. So, we adopt your suggestion, and replace “crustal remnants” by “Eoarchaean–Palaeoarchaean 'protocrust'”.

Revision: Line 21 “*primarily owing to the paucity of well-preserved Eoarchaean–Palaeoarchaean 'protocrust'.*”

2# Line 30 “*Further, zircon Lu–Hf isotopes reveal that these crustal fragments record Hadean to early Eoarchean (4.1–3.8 Ga) mantle extraction ages*”: **I would be a bit more cautious here...these 'ages' are fraught with uncertainty and highly model dependent.**

Response: We appreciate the reviewer’s concern about the model ages. This comment is along the lines of Comments 3 and 4. According to the reviewer’s suggestion, we have included a comprehensive discussion on the uncertainty of calculation of mantle model age, and present reasons why the two stage depleted mantle ages are suitable for these early Archean granitoids of this study—cited earlier in response to Comments 3 and 4. In addition, we replace “4.1–3.8 Ga” by “4.2–3.8 Ga” as suggested by the reprocessed T_{DM^2} ages. Please refer to Comment 3 and 4 for detailed discussion.

Revision: Line 24-25 “*ancient 4.2–3.8 Ga mantle extraction ages*”.

3#Line 57 “*Even in cratons that contain such Eoarchean–Paleoarchean rocks, they often only comprise a volumetrically minor component of the terrane itself*”: **Maybe true for**

Eoarchean rocks but, there are plenty of relatively well-preserved Paleoarchean rocks in the East Pilbara Terrane and Barberton area.

Response: We appreciate the reviewer for pointing out this ambiguous expression. It's true that Palaeoarchaeal rocks are more abundant compared to Eoarchaeal ones. We adopt your suggestion, and delete the reference to the Palaeoarchaeal here.

Revision: Lines 45-46 “Even in cratons that contain such Eoarchaeal rocks, they often only comprise a volumetrically minor component of the terrane itself¹².”.

4# Lines 98-100 “Generally, the two-stage depleted mantle model (T_{DM}) age of granitoids represents the timing of extraction of their mafic precursors from a depleted mantle source, which in turn signifies the timing of crustal growth.”: **Needs a reference.**

Revision: Good point. We added the following reference:

Mole, D.R. et al. (2019) Time-space evolution of an Archean craton: A Hf-isotope window into continent formation. *Earth-Science Reviews* 196, 102831.

5# Lines 101-102 “The 3.6–3.5 Ga zircon grains have sub-chondritic ϵ_{Hf} values ranging from -2.24 to -1.13 , with T_{DM2} ages of 4.1–3.8 Ga.”: **In Figure 2b is looks like the 3.8-4.2 Ga evolution array is defined by all the newly acquired data (red circles), not just the 3.5-3.6 Ga rocks. These older rocks appear to have a more limited range of zircon Hf compositions and hence model ages. I would prefer to see the range of models ages for each magmatic pulse separately and then provide a summarily at the end. In Figure 2b, the evolution array is defined by 3.8 and 4.2 Ga crustal evolution lines? Moreover, in Figure 2b it looks like there is a pulse of chondritic magmatism at c.3.8 Ga, that matches what is observed in the the North Atlantic Craton (Fig 2c). Why can't these rocks have formed by reworking this 3.8 Ga chondritic mafic crust?**

Response: We appreciate the reviewer's concern about presenting the results and discussion more logically. According to your suggestion, we have displayed the Hf isotopes for each magmatic pulse separately, and then give a summary. We agree with you that all the 3.6–3.5 Ga, 3.3–3.2 Ga and c. 2.72 Ga zircons have similar Hf T_{DM}^2 ages of 4.2–3.8 Ga, these analyses are well enveloped by 3.8 and 4.2 Ga crustal evolution lines in Fig. 2d. And we made some revisions. We partially agree with the reviewer that these 3.6–2.7 Ga zircons could be formed by reworking from a 3.8 Ga chondritic mafic crust, but only if a 3.8 Ga chondritic mantle existed in the North China Craton. However, as we discussed in Comment 3, several lines of evidence suggest a depleted mantle during the Eoarchaeal in the North China Craton. What we can confirm at this stage based on Hf isotopes is the 4.2–3.8 Ga mantle extraction age (i.e., crustal growth age) of the mafic protocrust. These 3.6–2.7 Ga zircon could be generated by the partial melting of mafic protocrust, which could be crystallized at any point between 4.2 and 3.8 Ga.

Revision: Lines 125-145 “The 3.6–3.5 Ga zircon grains from the Baishan nucleus have sub-chondritic $\epsilon_{Hf(t)}$ values ranging from -3.6 to -0.1 , with T_{DM}^2 ages of 4.2–3.9 Ga. In

contrast, the 3.3–3.2 Ga and c. 2.72 Ga groups display more unradiogenic Hf isotopic features with sub-chondritic $\epsilon_{\text{Hf}(t)}$ values ranging from –6.0 to –1.5 and –10.2 to –4.2, and both groups have similar T_{DM^2} ages of 4.2–3.8 Ga. Zircons from the c. 2.63 Ga monzogranite exhibit sub-chondritic $\epsilon_{\text{Hf}(t)}$ values ranging from –4.5 to –0.5 with T_{DM^2} ages of 3.7–3.3 Ga. Some of the 2.55–2.50 Ga potassic granites also exhibit unradiogenic Hf isotopes, yielding sub-chondritic $\epsilon_{\text{Hf}(t)}$ values ranging from –7.7 to –1.1, with T_{DM^2} ages of 3.7–3.3 Ga. In contrast, several 2.55–2.50 Ga potassic granites have more radiogenic Hf isotopes with positive $\epsilon_{\text{Hf}(t)}$ values ranging from 0 to +5.7 with T_{DM^2} ages of 3.2–2.8 Ga. Together, these data show that the 3.6–3.5, 3.3–3.2, and c. 2.72 Ga zircons have a similar T_{DM^2} age range (4.2–3.8 Ga), and they also display a common evolution as demonstrated by well-defined 4.2 and 3.8 Ga crustal evolution lines (**Fig. 2d**). The 4.2–3.8 Ga T_{DM^2} ages thus represent the mantle extraction age of the mafic protocrust, which subsequently experienced multiple stages of recycling at 3.6–3.5, 3.3–3.2, and c. 2.72 Ga. In addition, the Hf isotopes of the c. 2.78 Ga trondhjemitic gneiss¹³, the c. 2.63 Ga monzogranite, and the 2.55–2.50 Ga potassic granites with sub-chondritic $\epsilon_{\text{Hf}(t)}$ values reflect another mantle extraction event that occurred from 3.7–3.3 Ga (**Fig. 2d**). Moreover, the 2.55–2.50 Ga potassic granites with radiogenic Hf compositions, as well as the previously reported 2.55–2.50 Ga mafic volcanic rocks and TTGs from the Baishan nucleus (refs in **Supplementary Table S4**), suggest significant juvenile crustal growth between the Mesoarchaeon and Neoarchaeon. Together, these data demonstrate that the Baishan nucleus experienced multiple phases of crustal growth and reworking/recycling processes throughout the Archaean, leading to its geochemical maturation and facilitation of cratonization.”

6# “Analysis of zircon standard 91500 yielded a weighted mean $^{206}\text{Pb}/^{238}\text{U}$ age of 1062.3 ± 2.1 Ma, aligning with the recommended $^{206}\text{Pb}/^{238}\text{U}$ ages of 1064 Ma for this standard within analytical errors (Wiedenbeck et al., 1995). Analysis of Plešovice yielded a weighted mean $^{206}\text{Pb}/^{238}\text{U}$ age of 339.3 ± 4.1 Ma matched the recommended values (Sláma et al., 2008)”: **This is different to the date reported above? (1060.3 +- 2.9 Ma). I think its helpful to state what the accepted ages of the standards are so readers can compare for themselves.**

Response: We appreciate the reviewer for pointing out the lack of the analysis of standards. We have added the analyses results for all the standard materials into the Supplementary material. And we also added the accepted ages and values of standards for comparison.

Revision:

Lines 41-47 in Supplementary material “The standard 91500 (1062.4 ± 0.4 Ma; Wiedenbeck et al. ¹⁴) and NIST 610 were used for external calibration of the U–Pb ages and trace element content calculations respectively. Corrections for instrument drift, mass bias, and fractionation of the U–Pb ratio were made using a standard-sample bracketing method. For quality control purposes, the zircon standard Plešovice (337.13 ± 0.37 Ma, Sláma et al. ¹⁵) was analyzed after every ten unknown samples. Based on the actual situation, a testing denuded diameter of 30 μm was selected. The measured weighted

mean ages of standards 91500 (1060.4 ± 2.9 Ma) and Plešovice (339.6 ± 1.4 Ma) matched the recommended values.”

Lines 62-65 in Supplementary material “Analysis of zircon standard 91500 yielded a weighted mean $^{206}\text{Pb}/^{238}\text{U}$ age of 1062.2 ± 2.3 Ma, aligning with the recommended $^{206}\text{Pb}/^{238}\text{U}$ ages of 1062.4 ± 0.4 Ma for this standard within analytical errors¹⁴. Analysis of Plešovice yielded a weighted mean $^{206}\text{Pb}/^{238}\text{U}$ age of 335.9 ± 1.7 Ma matched the recommended values (337.13 ± 0.37 Ma; Sláma et al.¹⁵).”

Lines 82-84 in Supplementary material “The measured ages of zircon standards 91500 (weighted mean age = 1060.3 ± 5.0 Ma) and Plešovice (weighted mean age = 338.2 ± 3.9 Ma) are indistinguishable from accepted reference values suggested by Wiedenbeck et al.¹⁴ and Sláma et al.¹⁵.”

Lines 94-97 in Supplementary material “Standard zircon of TEMORA ($^{206}\text{Pb}/^{208}\text{U} = 416.7 \pm 1.3$ Ma)¹⁶ was utilized to correct the U, Th, and Pb contents as well as the ages of zircon. The measured ages of zircon standards TEMORA (weighted mean age = 417.3 ± 2.6 Ma) are consistent with the recommended age.”

Lines 129-133 in Supplementary material “Lu-Hf data for these zircon standards are listed in Table S3. The test $^{176}\text{Hf}/^{177}\text{Hf}$ values of Plešovice (weighted mean value = 0.282478), 91500 (weighted mean value = 0.282297), and GJ-1 (weighted mean value = 0.282005) (**Table S3**) were consistent with the recommended values within the error range (Plešovice = 0.282478 ± 8 , 91500 = 0.282300 ± 11 , and GJ-1 = 0.282009 ± 10) suggested by Zhang and Hu¹⁷.”

Lines 166-167 in Supplementary material “The analyses results of zircon standards TEMORA (weighted $\delta^{18}\text{O} = 8.22 \pm 0.11$ ‰) are consistent with the recommended value ($\delta^{18}\text{O} = 8.20$ ‰)¹⁶”.

7# “However, it’s worth noting that the mafic precursors of those earliest TTGs may had a long crustal residence time under a stagnant lid regime, potentially leading to significant underestimation of zircon ϵHf value and mantle depletion”: “**may have**”, **If there was a long crustal residence prior to remelting to form TTG then this should be reflected in the Hf isotope composition.**

Response: We appreciate the reviewer pointing out the error. We have replaced “may had” by “may have”. In order to better clarify the long crustal residence time, we compiled a new dataset of global igneous zircon Hf isotopes for Eoarchaeon to Mesoarchaeon TTGs from ancient cratons. The new compilation suggests that a considerable proportion of zircon display obviously low, sub-chondritic ϵHf values (Fig. 2c; see following figure). This indicates that the mafic precursors of these early Archaean TTGs may have a long crustal residence time, no matter whether their mafic precursors derived from a chondritic or depleted mantle (with both evolutionary trends depicted in the figure too).

Revision: Lines 103-108 “To directly address this issue, we compiled a global dataset of igneous zircon Hf isotopes of early Archaean TTGs from all major cratons on Earth (Table S4). The compilation reveals that most zircons display very low sub-chondritic $\epsilon_{\text{Hf}(t)}$ values (Fig. 2c), indicating that the mafic precursors of these early Archaean TTGs likely had a long crustal residence time, potentially leading to significant underestimation of zircon $\epsilon_{\text{Hf}(t)}$ value and the degree of mantle depletion¹.”

8# Is DM modelled as a straight line connecting chondrite at 4.5 Ga and present day compositions, or modelled using the starting composition at 4.5 Ga and $^{176}\text{Lu}/^{177}\text{Hf}$? Please provide more details.

Response: We appreciate the reviewer’s attention to the depleted mantle (DM) lines. The DM lines are based on the model of Griffin et al.¹⁸ with $^{176}\text{Hf}/^{177}\text{Hf}$ of 0.283251 and $^{176}\text{Lu}/^{177}\text{Hf}$ of 0.0384, and with depletion occurring at either 4.56 Ga or 3.8 Ga.

Revision: Lines 575-578 “Depleted mantle (DM) lines are based on models of a present depleted mantle with a $^{176}\text{Hf}/^{177}\text{Hf}$ of 0.283251 and a $^{176}\text{Lu}/^{177}\text{Hf}$ of 0.0384 (ref. ¹⁸), and the 3.8 Ga DM curve is extrapolated from the assumption that growth of the depleted mantle began at 3.8 Ga ($\epsilon_{\text{Hf}} = 0$) and evolved to the present-day depleted mantle reservoir³”.

9# “The weighted mean ϵ_{Hf} value and probability of fit (p) values of each sample were calculated using Origin 2023. Only samples with $p \geq 0.05$ are considered statistically homogeneous, while those with $p < 0.05$ are deemed heterogeneous and thus excluded. The compiled weighted mean ϵ_{Hf} values are presented in Table S4 and Fig. 2b, c.”: I’m not familiar with this software package. Could you provide a reference for it? I’m not sure this is appropriate. There are several reasons why zircon from a given rock might exhibit heterogeneous Hf isotope signatures. ‘Statistical homogeneity’

is not necessarily a good measure of data quality and my sense is that by filtering the data in such a manner you could be excluding real geological signals from the dataset. I think it would be better to filter out analyses whose U-Pb systematics have been affected by Pb loss (as shown in Figure S1 above). My sense is that if there is still heterogeneity in $^{176}\text{Hf}/^{177}\text{Hf}$ after this step, then it could indeed reflect real.

Response: We appreciate reviewer's questions about the data filtering process—and understand their concern about a heterogeneity being a potentially real effect. The Origin 2023 software includes basic data processing such as calculating means and standard variance, linear prediction, and hypothesis test (such as t -value test and variance test). Use of it is now cited in the Methods—although such stats calculations are not at all special to it. In term of statistical homogeneity, we have considered this comment carefully. According to the reviewer's suggestion, we firstly filter out igneous zircon with obvious Pb loss as well as metamorphic/recrystallized zircon. This step can remove those zircons that experienced intensive later metamorphism and/or hydrothermal alteration. However, zircon that experienced significant crustal contamination and/or mixing may or may not display any Pb loss. Thus, we still need another method for filtering them out. Generally, crustal contamination or magma mixing can induce heterogeneous Hf isotope signatures with widely variable ranges—as the reviewer points out. So, we adopt a probability of fit (p) values to filter out those analyses that have a wide range of Hf isotopes. So, yes, we indeed agree with the reviewer that such statistical filtering principle might exclude some real geological signals. So, we also plot these samples with $p < 0.05$, but in gray color on the diagrams and don't include them in the interpretation. Actually, precluding these statistical heterogeneous zircons (gray color in Fig. 2d, e) do not affect any interpretations.

Revision: Lines 384-388 “Generally, crustal contamination or magma mixing could induce heterogeneous Hf isotope signatures. Thus, this study applies probability of fit (p) values of each sample to filter out those zircons with heterogeneous Hf compositions. The probability of fit (p) values of each sample were calculated using Origin 2023 (www.originlab.com/2023). Only samples with $p \geq 0.05$ are considered statistically homogeneous, while those with $p < 0.05$ are deemed heterogeneous and thus excluded from interpretation.”

References

- Fisher, C. M., Vervoort, J. D., & Hanchar, J. M. (2014). Guidelines for reporting zircon Hf isotopic data by LA-MC-ICPMS and potential pitfalls in the interpretation of these data. *Chemical Geology*, 363, 125-133.
- Kemp, A. I., Vervoort, J. D., Petersson, A., Smithies, R. H., & Lu, Y. (2023). A linked evolution for granite-greenstone terranes of the Pilbara Craton from Nd and Hf isotopes, with implications for Archean continental growth. *Earth and Planetary Science Letters*, 601, 117895.

Petersson, A., Kemp, A. I., Hickman, A. H., Whitehouse, M. J., Martin, L., & Gray, C. M. (2019). A new 3.59 Ga magmatic suite and a chondritic source to the east Pilbara Craton. *Chemical Geology*, 511, 51-70.

Van Kranendonk, M. J., Smithies, R. H., Hickman, A. H., & Champion, D. C. (2007). Paleoproterozoic development of a continental nucleus: the East Pilbara terrane of the Pilbara craton, Western Australia. *Developments in Precambrian Geology*, 15, 307- 337.

Sincere thanks, for the critical and constructive comments.

Reviewer #2 (Remarks to the Author):

This manuscript presents new zircon U-Pb age, Lu-Hf isotope and Oxygen isotope data for the newly recognized 3.57 Ga Tonalite, together with other younger rocks, in the North China Craton. It represents the finding of a new continental nucleus in the North China Craton. Based on the data, the authors addressed three problems: the widespread of Hadean to early Eoarchean proto-crust, the geodynamic process for the Paleoproterozoic to Mesoproterozoic magmatism and the initiation of plate tectonics. The data is interesting and the manuscript is well organized. This research is generally of broad interests for researchers who are interested in the early earth. I suggest the authors add some new issues in the revision.

We are very encouraged that the reviewer finds the manuscript well-executed and of broad interest. We very much appreciate the issues raised to help improve during revision.

Comment 1: First, the readers would like to know if the Eoarchean to Paleoproterozoic rocks in the Jiapigou–Baishan Lake area of this study, and those reported in previous studies, mainly in the Anshan-Benxi area and the eastern Hebei area belong to the same coherent continental nucleus or three individual nuclei.

Response: Excellent question. Discussion on this new issue makes our manuscript more comprehensive, so the comment is greatly appreciated. Based on their similar ages of magmatic events and zircon Hf isotopes, these three nuclei might represent a coherent continental.

Revision: Lines 194-211 “The three nuclei within the North China Craton (i.e., Anshan, Baishan, and Eastern Hebei) are connected by Neoproterozoic granite–greenstone belts; however, it remains unclear whether these nuclei belonged to a single coherent continental terrane or represented three individual terranes. In favor of the former argument, we note that these nuclei experienced many similar Eoarchean-to-Mesoproterozoic magmatic events, such as the c. 3.8 Ga magmatic event recorded in the Anshan and Eastern Hebei nuclei¹⁹. The oldest c. 3.6 Ga magmatic event in the Baishan nucleus also correlates with that identified from the Hujiamiao Complex of the Anshan nucleus¹⁹. Also, intensive crustal reworking events occurred in all three nuclei during 3.3–2.9 Ga (refs. ²⁰⁻²²), leading to the formation of large-scale potassic granitoids and reflecting the existence of voluminous continental crust in each region during the

Mesoarchaean. Finally, zircon Hf isotopes suggest similar T_{DM}^2 ages (4.2–3.8 Ga), $^{176}\text{Lu}/^{177}\text{Hf}$ ratios (0.022), and crustal evolution histories (**Fig. 2d**) for these three nuclei. Therefore, the three North China nuclei may have once constituted a coherent Eoarchaean-to-Mesoarchaean proto-craton. Considering that c. 2.7–2.6 Ga potassic granites have only been identified from the Baishan nucleus, and the existence of a 2.7–2.6 Ga granite-greenstone belt between the Baishan and Anshan nuclei, we suggest that Baishan might have rifted away from Anshan during the early Neoarchaean, and then later reunited during late Neoarchaean cratonization of North China. Such a scenario involving rifting and breakup of an ancient continental nucleus resembles the processes that have been proposed for the Yilgarn Craton²³.
”

Comment 2: The other issue is about the geodynamic process of the 3.57-2.72 Ga magmatism. The authors proposed the 3.57-2.72 Ga magmatism, including tonalite, monzogranite and trondhjemite, were formed by repeated plume underplating in a stagnant lid background, mainly based on their Hf isotopes. This seems acceptable but not enough. The readers would wonder what controls the major and trace element geochemical characteristics (which is missing in the presented data) of the 3.57 Ga tonalite, 3.23 Ga monzogranite, 2.78 Ga trondhjemite, 2.72 Ga monzogranite if they are all derived from the same proto-crust. What are the differences in their melting conditions? What are the implications of such differences?

Response: We appreciate the question regarding the geochemistry. The geochemical data are provided in Supplementary Table S7. We have added discussion based on the major and trace elements for these granitoids, and provide related figures (Fig. 3, shown below). The 3.3–2.5 Ga potassic granites all exhibit geochemical features of adakitic granites derived from the partial melting of thickened crust with melting pressures >1.0 GPa and temperatures between 850 and 800°C. These 3.3–2.5 Ga adakitic granites reflect that the Baishan nucleus might maintain a thickened crust during 3.3–2.5 Ga.

Revision: Lines 147-173 “Geochemical constraints on petrogenesis of the 3.3–2.5 Ga potassic granites

In this study, a total of 20 samples were analyzed for major and trace elements. Detailed results of geochemistry can be found in the **Table S7**. The 3.3–2.5 Ga potassic granites exhibit similar geochemical features, such as high SiO_2 (70.38–77.94 wt. %), low MgO (0.18–1.21 wt. %), $^{\text{T}}\text{Fe}_2\text{O}_3$ (1.24–4.00 wt. %), Cr (3.16–26.4 ppm), and Ni (1.22–17.20 ppm) contents. They are enriched in light rare earth elements (e.g., La and Ce), Sr, Zr, and Hf, but depleted in heavy rare earth elements (e.g., Lu, Yb, and Y), Nb, and Ta (**Fig. 3a**), with mainly positive Eu anomalies. On a $\text{Al}_2\text{O}_3/(\text{FeO}^{\text{T}}+\text{MgO})-(3\text{CaO})-(5\text{K}_2\text{O}/\text{Na}_2\text{O})$ ternary diagram²⁴ (**Fig. 3b**), the 3.3–3.2 Ga and c. 2.72 Ga monzogranites plot mainly within the field of melts derived from tonalite. As the 3.3–3.2 Ga and c. 2.72 Ga zircons exhibit a similar Hf crustal evolutionary pattern to the 3.6–3.5 Ga xenocrystic zircons, it is reasonable to propose that the 3.6–3.5 Ga TTG was the source for the 3.3–3.2 Ga and c. 2.72 Ga monzogranites. The source rock for the c. 2.63 Ga monzogranite could have been a high-K mafic rock (**Fig. 3b**), which had a mantle extraction age of 3.7–3.3 Ga. The c. 2.5 Ga potassic granites exhibiting negative $\epsilon_{\text{Hf}(t)}$ values might have been

sourced from TTG rocks (**Fig. 3b**), whereas the other c. 2.5 Ga potassic granites displaying positive $\epsilon_{\text{Hf}(t)}$ values, higher Al_2O_3 values and $\text{K}_2\text{O}/\text{Na}_2\text{O}$ ratios, likely formed from juvenile metasediments. All of the 3.3–2.5 Ga potassic granites display high Sr/Y ratios (32–132, except for one analysis of 18) and $\text{La}_\text{N}/\text{Yb}_\text{N}$ (35–201, except for one analysis of 17), and can thus be classified as adakitic granites²⁵ (**Fig. 3c**). Generally, such adakitic geochemical characteristics suggest substantial garnet but minor plagioclase in the source region during partial melting, classically interpreted to occur at a pressure greater than 1.0 GPa (ref. ²⁵). Based on the computational method of Miller et al. ²⁶, the average Zr saturation temperatures (T_{Zr}) of the 3.3–3.2, c. 2.72, c. 2.63, and c. 2.5 Ga adakitic granites are 794 °C, 847 °C, 818 °C, and 808 °C, respectively. Moreover, the characteristics of high SiO_2 contents but low MgO, Cr, and Ni contents suggest a thickened crust origin for these adakitic granites²⁷ (**Fig. 3d**). This interpretation is also supported by abundant xenocrystic zircons within the 3.3–2.5 Ga potassic granites and their unradiogenic zircon Hf isotopes (e.g., negative zircon $\epsilon_{\text{Hf}(t)}$ values and Hadean to Palaeoarchaeon T_{DM^2} ages). As such, these adakitic granites indicate that the Baishan nucleus maintained a notably thick continental crust (>30 km) from at least 3.3 Ga to 2.5 Ga.”

Fig. 3

Comment 3: And I disagree with the authors on their discussion on the onset of plate tectonics. The authors proposed asynchronous onset of plate tectonics in different cratons based on asynchronous increase of Hf isotope and O isotopes. In my opinion, the asynchronous onset of plate tectonics is a fake proposition because plate tectonics is defined to be a global tectonics characterized by the creation and maintenance of a global network of narrow plate boundaries. The increase of Hf isotope ratios may signify the contribution of depleted mantle related to plate subduction. The increase of O isotope ratios suggests the emergence of large-scale subaerial landmass for low-temperature weathering.

Response: We appreciate the reviewer for clarifying the asynchronous onset of subduction rather than the asynchronous onset of plate tectonics. We agree with you that only the emergence of global network of plate boundaries represents the beginning of plate tectonics. Asynchronous onset of subduction in different terranes would promote the onset of the global network of plate boundaries. Therefore, we change “the onset of plate tectonics” to “the onset of subduction” throughout the manuscript. We accept that asynchronous increases of $\delta^{18}\text{O}$ values in different cratons suggests the asynchronous emergence of large-scale subaerial land. To illustrate the role of subduction during the late Neoproterozoic in the NCC, we also consider other lines of geological evidence, such as ca. 2.5 Ga paired metamorphism in Dengfeng²⁸, Alpine-style sub-horizontal arc-affinity nappe structures in central NCC²⁹⁻³¹, and the widespread presence of subduction-related potassic granites and sanukitoids³². Therefore, we consider that the significant elevation of the $\delta^{18}\text{O}$ values of the ca. 2.5 Ga igneous rocks from the NCC was most likely caused by subduction that incorporated large volumes of supracrustal materials into magma sources. Compared to other Archean cratons, the records of intensive subduction in the NCC occur later than those in other cratons, which is therefore consistent with the interpretation that the onset of subduction was asynchronous from a global perspective.

Revisions:

Line 271 “Archaean supercratons and the asynchronous onset of subduction”

Lines 303-326 “Globally, >2.7 Ga zircons primarily display mantle-like or slightly elevated $\delta^{18}\text{O}$ values, whereas those with ages <2.7 Ga show an increasing trend in $\delta^{18}\text{O}$ values (Fig. 2f), as similarly demonstrated in previous analyses³³. In North China specifically, zircon $\delta^{18}\text{O}$ values only begin to show significant elevation by the end of the Neoproterozoic (c. 2.5 Ga), while 3.6–2.6 Ga zircon exhibit mantle-like or only slightly elevated $\delta^{18}\text{O}$ values (Fig. 2f). Significant elevation of $\delta^{18}\text{O}$ values during the Neoproterozoic is consistent with the observation from triple-oxygen-isotopes recorded in Archaean shales³⁴ that reflect substantial emergence of subaerial landmass, allowing for subaerial weathering. The appearance of more exposed continent during the Neoproterozoic is contemporaneous with the assembly of the Kenorland supercontinent³⁵ or multiple supercratons³⁶ (i.e., Superia, Sclavia, and Vaalbara). Thus, the diachronous increase of $\delta^{18}\text{O}$ values in different cratons at different times in the Neoproterozoic suggests the asynchronous emergence of large-scale subaerial land.

There is broad consensus that plate tectonics had become established on Earth by around 3 Ga (refs. ³⁷⁻³⁹), even if localized subduction began in some cratons at an earlier time⁴⁰⁻⁴². The $\delta^{18}\text{O}$ values of the c. 2.5 Ga igneous rocks from North China show notably

higher values than those in older zircon (**Fig. 2f**), which suggest that considerable volumes of supracrustal materials have been incorporated into magma source regions. This process can be achieved through several geodynamic processes (e.g., subduction, sagduction, or thrust stacking), of which, subduction is considered the most effective way. Several independent lines of evidence confirm that subduction initiated locally within the North China Craton by the end of the Neoarchaeon, including evidence of paired metamorphism in Dengfeng²⁸, Alpine-style subhorizontal arc-affinity nappe structures in central North China^{29, 30}, and the widespread presence of subduction-related potassic granites and sanukitoids³²—all occurring at c. 2.5 Ga. Therefore, the initiation of subduction in North China might have occurred notably later than in many other cratons. Our new data thus add to a growing set of observations supporting the onset of subduction being asynchronous from a global perspective^{40, 43}.

Below are detailed comments and suggestions.

Comment 4: Line 121: The title is not consistent with the contents of this section. I can only see the crustal architecture of northeastern NCC from this paragraph. I can see nothing about the structure of lithospheric mantle in the whole manuscript.

Response: Fair point, thanks for pointing this out. We agree with the reviewer that this section is about the crustal structure rather than the structure of the lithospheric mantle. In addition, after clarifying the crustal architecture, we think that it is suitable for deciphering more about the affinities among the Baishan, Anshan, and Eastern Hebei nuclei here.

Revision: The revised section title (**Line 175**) is: “**Crustal architecture of northeastern North China and affinities among nuclei within the craton**”.

Comment 5: Lines 127-128: You need to provide the ages of these so-called “old crust”. Only unradiogenic $\epsilon\text{Hf}(t)$ values do not support they are very old.

Response: Thanks for pointing out this unclear expression. Beside the ancient T_{DM^2} ages (3.7–3.3 Ga), these two regions also have some geological records that support that they are old than the surrounding Neoarchaeon granite–greenstone belt: for example, the c. 3.1 Ga TTG and amphibolite assemblage in the North Qingyuan area⁴⁴, and abundant 2.9–2.7 Ga xenocrystic zircon within the ca. 2.7 Ga meta-mafic volcanic rocks⁴⁵. Thus, we think these two regions might represent two reworked ancient crustal fragments.

Revision: Lines 183-187 “Additionally, two small regions in the Qingyuan and Helong areas display ancient T_{DM^2} ages of 3.8–3.3 Ga and unradiogenic Hf isotopes, probably indicating the existence of ancient crustal fragments (**Fig. 4**). This interpretation is further supported by the identification of the c. 3.1 Ga TTG and amphibolite assemblage⁴⁴ and abundant 2.9–2.7 Ga xenocrystic zircons within the c. 2.7 Ga meta-mafic volcanic rocks⁴⁵.”

Comment 6: The effective numbers: For example, the uncertainty of $\epsilon\text{Hf}(t)$ values is

about 0.4-0.5. Therefore, the significant digit after the decimal point of $\epsilon\text{Hf}(t)$ should be 1. And the significant digit of $^{176}\text{Hf}/^{177}\text{Hf}$ and error is traditionally 6.

Revision: Good point about sig figs. We revised this throughout the main text and supplementary materials and tables.

Comment 7: Fig. 3a: The design of this figure is not logical. The hot and cold color is designed to represent negative and positive $\epsilon\text{Hf}(t)$ values. However, the same $\epsilon\text{Hf}(t)$ values at different ages represent different mantle extraction age. If they are labelled as the same color, the authors would get the wrong information that the mantle extraction ages are the same. I strongly suggest the authors provide zircon U-Pb age map and T_{DM^2} map in Fig. 3.

Revision: Good suggestion for better portraying the contour maps. Providing a zircon U-Pb age contour map is helpful to clarify the crustal architecture of this region. We assume that the reviewer might have missed the Hf T_{DM^2} map in the previous version. So, in the revision, we added a new zircon U-Pb age map and also added titles for each of the contour maps to help readers easily understand. **Revised Figure 4 shown below:**

Fig. 4

Comment 8: Lines 146-148: It is impossible for the existence of a “global-scale” Hadean to early Eoarchean proto-crust. This means that almost all continental crust were formed in the very early stage of the Earth history. If so, most of the continental crust should have modal age of Hadean to early Eoarchean, which was not observed in zircon Hf model

age data base (e.g., Dhuime et al., 2012).

Dhuime, B., Hawkesworth, C.J., Cawood, P.A., Storey, C.D., 2012. A change in the geodynamics of continental growth 3 billion years ago. *Science* 335, 1334-1336.

Response: We appreciate the reviewer for pointing out this inappropriate expression. We agree with the reviewer that it cannot be true that almost all continental crust were formed in the Hadean to early Eoarchaeon. Our original view was that the Hadean to early Eoarchaeon protocrust might have existed widely within those oldest continental nuclei, such as the Slave, Yilgarn, and North China cratons. Therefore, we changed the title of this section to “**Hadean to early Eoarchaeon mafic protocrust in early Earth**”, and revised this section accordingly.

Revision: Lines 213-232 “Hadean to early Eoarchaeon mafic protocrust in early Earth

As shown by our new data, zircon Hf isotopes indicate that the Anshan, Baishan, and Eastern Hebei nuclei originated from a Hadean to early Eoarchaeon mafic protocrust (**Fig. 2d**). ^{142}Nd and ^{143}Nd isotopes for the 3.8–3.0 Ga Anshan Complex suggest 4.5–4.4 Ga model ages for the precursor of the oldest components and multiple mantle-crust differentiation events from 4.3 to 3.8 Ga (ref. ¹⁰). Several Hadean detrital zircons¹⁹ and xenocrystic zircons⁴⁶ identified from the Eastern Hebei and Anshan nuclei and the southern margin of North China provide direct evidence of a Hadean to early Eoarchaeon heritage for the craton. This interpretation is further corroborated by the Xinyang Eoarchaeon xenoliths from southern North China⁴⁷ that display more evolved Hf isotopes than the other three northern continental nuclei (**Fig. 2d**). Our new compilation of zircon Hf isotopes of Eoarchaeon-to-Mesoarchaeon TTGs worldwide show that the vast majority of Eoarchaeon–Palaeoarchaeon TTGs from most cratons (e.g., Slave, Yilgarn, East Antarctica, Superior, and Kaapvaal) generally display T_{DM^2} ages >4 Ga (**Fig. 2c**). Studies of the Eoarchaeon Acasta Gneiss Complex of the Slave Craton suggest that these oldest rocks on Earth were generated from a Hadean mafic protocrust^{12, 48}. In light of these earliest records and our new findings from North China Craton, we propose the existence of a Hadean to early Eoarchaeon mafic protocrust that was crucial for the formation of the earliest continental crust nuclei preserved in individual cratons⁴⁹.”

Comment 9: Lines 152-154: The statement is not consistent with Fig. 2b. Not all the >2.6 Ga granitoids within the ABN exhibit a simple linear evolutionary ϵHf value trend. The 3.1-3.0Ga rocks in the ABN show obvious increase in ϵHf values.

Response: We appreciate the reviewer for pointing out this ambiguous description. According to the zircon Hf isotopes, the ancient nuclei of the NCC might preserve 4.2–3.8 Ga mafic protocrust, and younger 3.8–3.3 Ga later underplated mafic crust. Thus, two different episodes of melting of these two different mafic sources would have two crustal evolutionary trends, as illustrated in the new Fig. 2d shown below. We have also revised this ambiguous description in the main text.

Fig. 2d

Revision: Lines 233-242 “The 3.8–3.6 Ga TTG and the c. 3.3 and 3.1–2.9 Ga potassic granites in the Anshan nucleus, as well as the 3.3–2.7 Ga granitoids within the Baishan nucleus all exhibit consistent linear crustal evolutionary trends (melting trend #1 in **Fig. 2d**), and were therefore derived from repeated melting of a 4.2–3.8 Ga mafic precursor(s), without significant addition of juvenile materials in any magmatic episode. In addition, the c. 3.45 Ga migmatite, the c. 3.3 Ga trondhjemite, and the c. 3.1 Ga trondhjemite from the Anshan nucleus, the c. 2.9 Ga TTG and diorite from the Eastern Hebei nucleus, as well as the c. 2.78 Ga trondhjemite, c. 2.63 monzogranite, and some 2.5 Ga potassic granites from the Baishan nucleus define another reworking episode with a distinct crustal evolution array (melting trend #2 in **Fig. 2d**). They represent repeated melting products of former (3.8–3.3 Ga) underplated mafic crusts.”

Comment 10: Lines 160-161: I cannot see the crustal architecture characterized by “younger and juvenile granite-greenstone belts surrounding ancient, long-lived, and reworked continental nuclei” from Fig. 3.

Response: Thanks for pointing out this ambiguity. We assume that maybe due to the modification by younger orogenic belts, such an original crustal architecture might not be easy to recognize. Nonetheless, we would like to insist on our view. The Baishan nucleus and the two other ancient crustal fragments are distributed as isolated “fragment” on the northern margin of the NCC, and are separated from the Anshan nucleus by the late Neoproterozoic North Liaoning to South Jilin granite–greenstone belt (new Fig. 4). The zircon T_{DM}^2 age contour map might be a clearer way to see that these nuclei (warm color) are

surrounded by the younger and juvenile granite–greenstone belt (cold color). We sketched approximately such architecture in the following figure for illustration.

Revision: Lines 187-191 “The Baishan nucleus and the other two ancient crustal fragments occur as isolated fragments on the northern margin of the North China Craton, and are separated from the Anshan nucleus and each other by the late Neoproterozoic North Liaoning to South Jilin granite–greenstone belt (**Fig. 4**). This crustal architecture suggests that these ancient nuclei are surrounded by a younger and juvenile granite–greenstone belt.”

Comment 11: Lines 195-206: I personally agree that the NCC may belong to the Sclavia supercraton. But I cannot agree with the logics of the first evidence. The identification of Hadean-Eoarchean rocks and detrital zircons in these cratons, do not mean that they are in the same supercraton in the Neoproterozoic. The Hadean detrital zircons might be transported to other cratons at any time if they were in the same supercontinent.

Response: We appreciate the reviewer their concern about the logic of this evidence. We agree with the reviewer that if these cratons belonged to the same ancestral landmass (supercontinent or supercraton), detrital zircons with similar ages might be transported from other cratons. Thus, for this first potential line of evidence, we try to express it in a more logical way. These cratons recorded similar Hadean–Eoarchean tectono-thermal events, as evidenced by U–Pb ages and Hf isotopes of igneous zircon (Fig. 2c, d) from the Acasta complex of the Slave Craton^{12, 50}, the Narryer terrane of the Yilgarn Craton⁵¹, the Anshan and Eastern Hebei nuclei of the NCC, as well as the Mairi gneiss complex of the São Francisco Craton⁵².

Revision: Lines 285-289 “(i) similar Hadean–Eoarchean tectono-thermal events, as evidenced by U–Pb ages and Hf isotopes of igneous zircons (**Fig. 2c, d**) from the Acasta complex in the Slave Craton^{12, 50}, the Narryer terrane in the Yilgarn Craton⁵¹, the Anshan and Eastern Hebei nuclei in North China, as well as the Mairi gneiss complex in the São Francisco Craton⁵²,”

Comment 12: Lines 213-215: Subduction is not a pre-requirement for the increase of

$\delta^{18}\text{O}$ values. Other geodynamic processes for recycling of supracrustal materials (e.g., sagduction) are also capable to incorporate supracrustal materials into magmatic sources. The key point is the emergence of subaerial landmass for low-temperature weathering. Therefore, the asynchronous increase of $\delta^{18}\text{O}$ values in different cratons suggest the asynchronous emergence of large-scale subaerial land rather than asynchronous initiation of subduction.

Bindeman, I.N., Zakharov, D.O., Palandri, J., Greber, N.D., Dauphas, N., Retallack, G.J., Hofmann, A., Lackey, J.S., Bekker, A., 2018. Rapid emergence of subaerial landmasses and onset of a modern hydrologic cycle 2.5 billion years ago. *Nature* 557, 545-548.

Response: We appreciate the reviewer for clarifying the meaning of the increase in $\delta^{18}\text{O}$ values. We accept that the asynchronous increase of $\delta^{18}\text{O}$ values in different cratons suggest the asynchronous emergence of large-scale subaerial land. To illustrate the role of subduction during the late Neoproterozoic in the NCC, we also consider other lines of geological evidence, such as ca. 2.5 Ga paired metamorphism in Dengfeng²⁸, Alpine-style sub-horizontal arc-affinity nappe structures in central NCC²⁹⁻³¹, and the widespread presence of subduction-related potassic granites and sanukitoids³². Therefore, we consider that the significant elevation of the $\delta^{18}\text{O}$ values of the ca. 2.5 Ga igneous rocks from the NCC was most likely caused by subduction that incorporated large volumes of supracrustal materials into magma sources. Compared to other Archean cratons, the records of intensive subduction in the NCC occur later than those in other cratons, which is therefore consistent with the interpretation that the onset of subduction was asynchronous from a global perspective.

Revision: Lines 303-329 “Globally, >2.7 Ga zircons primarily display mantle-like or slightly elevated $\delta^{18}\text{O}$ values, whereas those with ages <2.7 Ga show an increasing trend in $\delta^{18}\text{O}$ values (**Fig. 2f**), as similarly demonstrated in previous analyses³³. In North China specifically, zircon $\delta^{18}\text{O}$ values only begin to show significant elevation by the end of the Neoproterozoic (c. 2.5 Ga), while 3.6–2.6 Ga zircon exhibit mantle-like or only slightly elevated $\delta^{18}\text{O}$ values (**Fig. 2f**). Significant elevation of $\delta^{18}\text{O}$ values during the Neoproterozoic is consistent with the observation from triple-oxygen-isotopes recorded in Archean shales³⁴ that reflect substantial emergence of subaerial landmass, allowing for subaerial weathering. The appearance of more exposed continent during the Neoproterozoic is contemporaneous with the assembly of the Kenorland supercontinent³⁵ or multiple supercratons³⁶ (i.e., Superia, Sclavia, and Vaalbara). Thus, the diachronous increase of $\delta^{18}\text{O}$ values in different cratons at different times in the Neoproterozoic suggests the asynchronous emergence of large-scale subaerial land.

There is broad consensus that plate tectonics had become established on Earth by around 3 Ga (refs. ³⁷⁻³⁹), even if localized subduction began in some cratons at an earlier time⁴⁰⁻⁴². The $\delta^{18}\text{O}$ values of the c. 2.5 Ga igneous rocks from North China show notably higher values than those in older zircon (**Fig. 2f**), which suggest that considerable volumes of supracrustal materials have been incorporated into magma source regions. This process can be achieved through several geodynamic processes (e.g., subduction, sagduction, or thrust stacking), of which, subduction is considered the most effective way. Several independent lines of evidence confirm that subduction initiated locally within the North China Craton by the end of the Neoproterozoic, including evidence of paired

metamorphism in Dengfeng²⁸, Alpine-style subhorizontal arc-affinity nappe structures in central North China^{29, 30}, and the widespread presence of subduction-related potassic granites and sanukitoids³²—all occurring at c. 2.5 Ga. Therefore, the initiation of subduction in North China might have occurred notably later than in many other cratons. Our new data thus add to a growing set of observations supporting the onset of subduction being asynchronous from a global perspective^{40, 43}.”

Comment 13: Line 225: onset of plate tectonics or onset of subduction? Plate tectonics is defined to be a global tectonics characterized by the creation and maintenance of a global network of narrow plate boundaries. According to its definition, if plate tectonics began, it is globally. Therefore, you can say asynchronous onset of subduction but cannot say asynchronous onset of plate tectonics.

Response: We appreciate the reviewer for clarifying the asynchronous onset of subduction rather than the asynchronous onset of plate tectonics. We agree with you that only the emergence of global network of plate boundaries represents the *bona fide* beginning of plate tectonics *sensu strictu*. Only the progressive propagation of subduction initiation in different terranes would eventually promote the onset of the global network of plate boundaries^{40, 53}.

Revision: Lines 29, 271, 326 and 331 “onset of subduction”

Comment 14: GPS coordinates should be provided in the supplementary tables.

Response: Thanks for pointing out the omission of the GPS coordinates, we have provided GPS coordinates in the Supplementary Tables.

Comment 15: The age of 22BS20-3: From the field photo of Fig. S3b and d, I cannot see any difference between the 3.57 Ga tonalite and the 3.2 Ga monzogranite. There are three groups of ages in Fig. S6a and b, which group really record the crystallization age? To support your interpretation that the 3.57 Ga group is the crystallization age, you need to provide more zircon CL images and describe their Th/U ratios.

Response: We appreciate the reviewer’s concern regarding the interpretation of these two groups of ages. We reevaluated the “3.57 Ga” tonalite, where both c. 3.57 Ga and c. 3.3 Ga zircon groups from sample 22BS20-3 display concentric oscillatory zones with narrow 2.5 Ga recrystallized zones or metamorphic rims (**Fig. 2b**). Both the c. 3.57 Ga and c. 3.3 Ga zircon groups have high Th/U ratios (0.11–0.81; except three analyses of 0.01, 0.07, and 0.08). We note that sample 22BS20-3 is a small xenolith (approximately 20 x 30 cm), and its zircon U–Pb ages are similar to the surrounding c. 3.23 Ga monzogranites. Although we haven’t found any c. 3.3 Ga zircons with 3.57 Ga zircon cores, it might be more acceptable that the c. 3.3 Ga group represents the crystallization age, whereas the 3.57 Ga group merely represents inherited zircons. It is worth noting that larger amounts of ca. 3.6–3.5 Ga xenocrystic zircons existed in the Baishan nucleus, and the younger c. 3.3 Ga and c. 2.73 Ga granites show similar Hf crustal evolution trends with these 3.6–3.5 Ga zircon. These lines of evidence also allow us to draw a conclusion

that the oldest magmatic event of this region occurred at 3.6–3.5 Ga. Meanwhile, the other major conclusions are not affected if sample 22BS20-3 formed at c. 3.3 Ga with abundant 3.6–3.5 Ga xenocrystic zircon.

Revision:

Lines 71-76 “An early Palaeoarchaean magmatic event is recorded by abundant 3.6–3.5 Ga xenocrystic zircons found within younger (3.3–3.2 and 2.55–2.50 Ga) granitoids. These xenocrysts exhibit concentric oscillatory internal zonation, suggestive of crystallization from a felsic melt (**Fig. 2b**). Evidence for a subsequent late Palaeoarchaean/early Mesoarchaean magmatic event comes from 3.3–3.2 Ga monzogranites and 3.3–3.2 Ga xenocrystic zircons within younger 2.55–2.50 Ga potassic granitoids.”

Line 177 in Supplementary Materials “2.1 c. 3.3 Ga monzogranite (22BS20-3) with 3.6–3.5 Ga xenocrystic zircons”.

Lines 185-194 in Supplementary Materials “The cores display concentric oscillatory zones with high Th/U ratios (0.11–0.81, except three analyses of 0.01, 0.07, and 0.08) suggestive of a magmatic origin. Conversely, zircon rims and mantles are relatively luminescent or lack interior structures, indicating a metamorphic origin. Analyses on zircon cores can be divided into two groups. The older age group dated by LA-ICP-MS and SHRIMP yield $^{207}\text{Pb}/^{206}\text{Pb}$ ages of 3570–3364 Ma and 3573–3283 Ma with upper intercept ages of 3558±61 Ma and 3546±20 Ma, respectively. The four concordant analyses conducted by SHRIMP yielded a weighted mean age of 3571 ± 2 Ma (MSWD=1.06), which is interpreted as the age of xenocrystic zircons. The younger age group dated by LA-ICP-MS and SHRIMP yield $^{207}\text{Pb}/^{206}\text{Pb}$ ages of 3239–3076 Ma and 3252–2917 Ma with upper intercept ages of 3271±25 Ma and 3238±45 Ma, respectively. This younger ca. 3.3 Ga is interpreted as the crystallization age for Sample 22BS20-3.”

Fig. 2b

Comment 16: Lu-Hf isotope and Oxygen isotope results for the metamorphic domains should not be plotted on the figures. If the zircon metamorphic domain is of

recrystallization origin, its $^{176}\text{Hf}/^{177}\text{Hf}$ isotope ratios should be similar with its igneous predecessor whereas its U-Pb age is reset. If the Hf isotope is plotted on the $\epsilon\text{Hf}(t)$ -age diagram, it will mislead the readers that the magmatism of this age have such $\epsilon\text{Hf}(t)$ values.

Response: We appreciate the reviewer for reminding us not to plot the results of metamorphic domains. According to your suggestion, we exclude the Hf and O isotopic data from metamorphic or recrystallized domains before plotting.

Reference

1. Liou, P. et al. Zircons underestimate mantle depletion of early Earth. *Geochimica et Cosmochimica Acta* **317**, 538-551 (2022).
2. Fisher, C.M. & Vervoort, J.D. Using the magmatic record to constrain the growth of continental crust—The Eoarchean zircon Hf record of Greenland. *Earth and Planetary Science Letters* **488**, 79-91 (2018).
3. Ge, R. et al. Generation of Eoarchean continental crust from altered mafic rocks derived from a chondritic mantle: The ~3.72 Ga Aktash gneisses, Tarim Craton (NW China). *Earth and Planetary Science Letters* **538** (2020).
4. Petersson, A. et al. A new 3.59 Ga magmatic suite and a chondritic source to the east Pilbara Craton. *Chemical Geology* **511**, 51-70 (2019).
5. Kemp, A.I.S., Vervoort, J.D., Petersson, A., Smithies, R.H. & Lu, Y. A linked evolution for granite-greenstone terranes of the Pilbara Craton from Nd and Hf isotopes, with implications for Archean continental growth. *Earth and Planetary Science Letters* **601** (2023).
6. Blichert-Toft, J., Arndt, N.T. & Gruau, G. Hf isotopic measurements on Barberton komatiites: effects of incomplete sample dissolution and importance for primary and secondary magmatic signatures. *Chemical Geology* **207**, 261-275 (2004).
7. Nebel, O., Campbell, I.H., Sossi, P.A. & Van Kranendonk, M.J. Hafnium and iron isotopes in early Archean komatiites record a plume-driven convection cycle in the Hadean Earth. *Earth and Planetary Science Letters* **397**, 111-120 (2014).
8. Jayananda, M., Guitreau, M., Aadhiseshan, K.R., Miyazaki, T. & Chung, S.L. Origin of the oldest (3600–3200 Ma) cratonic core in the Western Dharwar Craton, Southern India: Implications for evolving tectonics of the Archean Earth. *Earth-Science Reviews* **236** (2023).
9. Mitra, A., Dey, S., Zong, K., Liu, Y. & Mitra, A. Building the core of a Paleoproterozoic continent: Evidence from granitoids of Singhbhum Craton, eastern India. *Precambrian Research* **335** (2019).
10. Li, C.-F. et al. Differentiation of the early silicate Earth as recorded by ^{142}Nd - ^{143}Nd in 3.8–3.0 Ga rocks from the Anshan Complex, North China Craton. *Precambrian Research* **301**, 86-101 (2017).
11. Amelin, Y., Lee, D., Halliday, A.N. & Pidgeon, R.T. Nature of the Earth's earliest crust from hafnium isotopes in single detrital zircons. *Nature* **399**, 252-255 (1999).
12. Reimink, J.R. et al. No evidence for Hadean continental crust within Earth's oldest evolved rock unit. *Nature Geoscience* **9**, 777-780 (2016).

13. Wu, M., Lin, S., Wan, Y., Gao, J.-F. & Stern, R.A. Episodic Archean crustal accretion in the North China Craton: Insights from integrated zircon U-Pb-Hf-O isotopes of the Southern Jilin Complex, northeast China. *Precambrian Research* **358**, 106150 (2021).
14. Wiedenbeck, M. et al. Three Natural Zircon Standards for U-Th-Pb, Lu-Hf, Trace Element and Ree Analyses. *Geostandards and Geoanalytical Research* **19**, 1-23 (1995).
15. Sláma, J. et al. Plešovice zircon — A new natural reference material for U–Pb and Hf isotopic microanalysis. *Chemical Geology* **249**, 1-35 (2008).
16. Black, L.P. et al. Improved $^{206}\text{Pb}/^{238}\text{U}$ microprobe geochronology by the monitoring of a trace-element-related matrix effect; SHRIMP, ID–TIMS, ELA–ICP–MS and oxygen isotope documentation for a series of zircon standards. *Chemical Geology* **205**, 115-140 (2004).
17. Zhang, W. & Hu, Z. Estimation of Isotopic Reference Values for Pure Materials and Geological Reference Materials. *Atomic Spectroscopy* **41**, 93-102 (2020).
18. Griffin, W.L. et al. Zircon chemistry and magma mixing, SE China: In-situ analysis of Hf isotopes, Tonglu and Pingtan igneous complexes. *Lithos* **61**, 237-269 (2002).
19. Wan, Y. et al. Hadean to early Mesoarchean rocks and zircons in the North China Craton: A review. *Earth-Science Reviews* **243**, 104489 (2023).
20. Dong, C. et al. The Mesoarchean Tiejiaoshan-Gongchangling potassic granite in the Anshan-Benxi area, North China Craton: Origin by recycling of Paleo- to Eoarchean crust from U-Pb-Nd-Hf-O isotopic studies. *Lithos* **290-291**, 116-135 (2017).
21. Liou, P. & Guo, J. Deciphering the Mesoarchean to Neoproterozoic history of crustal growth and recycling in the Caochang region of the Eastern Hebei Province, North China Craton using combined zircon U–Pb and Lu-Hf isotope analysis. *Lithos* **334-335**, 281-294 (2019).
22. Wang, Y.F., Li, X.H., Jin, W., Zeng, L. & Zhang, J.H. Generation and maturation of Mesoarchean continental crust in the Anshan Complex, North China Craton. *Precambrian Research* **341** (2020).
23. Mole, D.R. et al. Time-space evolution of an Archean craton: A Hf-isotope window into continent formation. *Earth-Science Reviews* **196**, 102831 (2019).
24. Laurent, O., Martin, H., Moyen, J.F. & Doucelance, R. The diversity and evolution of late-Archean granitoids: Evidence for the onset of “modern-style” plate tectonics between 3.0 and 2.5Ga. *Lithos* **205**, 208-235 (2014).
25. Drummond, M.S. & Defant, M.J. A model for Trondhjemite-Tonalite-Dacite Genesis and crustal growth via slab melting: Archean to modern comparisons. *Journal of Geophysical Research* **95**, 21503-21521 (1990).
26. Miller, C.F., McDowell, S.M. & Mapes, R.W. Hot and cold granites? Implications of zircon saturation temperatures and preservation of inheritance. *Geology* **31**, 529-532 (2003).
27. Wang, Q. et al. Petrogenesis of Adakitic Porphyries in an Extensional Tectonic Setting, Dexing, South China: Implications for the Genesis of Porphyry Copper Mineralization. *Journal of Petrology* **47**, 119-144 (2006).
28. Huang, B. et al. Paired metamorphism in the Neoproterozoic: A record of accretionary-to-collisional orogenesis in the North China Craton. *Earth and Planetary Science Letters* **543**, 116355 (2020).
29. Zhong, Y. et al. Alpine-style nappes thrust over ancient North China continental margin demonstrate large Archean horizontal plate motions. *Nat Commun* **12**, 6172 (2021).
30. Zhong, Y., Kusky, T.M. & Wang, L. Giant sheath-folded nappe stack demonstrates extreme subhorizontal shear strain in an Archean orogen. *Geology* **50**, 577-582 (2022).

31. Zhong, Y. et al. Alpine-style tectonic nappe stacking in an Archean suture zone: Quantitative structural profile places constraints on orogenic architecture. *Gondwana Research* **117**, 86-116 (2023).
32. Wan, Y. et al. Zircon ages and geochemistry of late Neoproterozoic syenogranites in the North China Craton: A review. *Precambrian Research* **222-223**, 265-289 (2012).
33. Mitchell, R.N. et al. The supercontinent cycle. *Nature Reviews Earth & Environment* **2**, 358-374 (2021).
34. Bindeman, I.N. et al. Rapid emergence of subaerial landmasses and onset of a modern hydrologic cycle 2.5 billion years ago. *Nature* **557**, 545-548 (2018).
35. Williams, H., Hoffman, P.F., Lewry, J.G., Monger, J.W.H. & River, T. Anatomy of North America: thematic geologic portrayals of the continent. *Tectonophysics* **187**, 117-134 (1991).
36. Bleeker, W. The late Archean record: a puzzle in ca. 35 pieces. *Lithos* **71**, 99-134 (2003).
37. Dhuime, B., Hawkesworth, C., Cawood, P.A. & Storey, C.D. A Change in the Geodynamics of Continental Growth 3 Billion Years Ago. *Science* **335**, 1334-1336 (2012).
38. Palin, R.M. et al. Secular change and the onset of plate tectonics on Earth. *Earth-Science Reviews* **207**, 103172 (2020).
39. Tang, M., Chen, K. & Rudnick, R.L. Archean upper crust transition from mafic to felsic marks the onset of plate tectonics. *Science* **351**, 372-375 (2016).
40. Huang, G., Mitchell, R.N., Palin, R.M., Spencer, C.J. & Guo, J. Barium content of Archean continental crust reveals the onset of subduction was not global. *Nat Commun* **13**, 6553 (2022).
41. Mitchell, R.N., Spencer, C.J., Kirscher, U. & Wilde, S.A. Plate tectonic-like cycles since the Hadean: Initiated or inherited? *Geology* **50**, 827-831 (2022).
42. Zhang, Q. et al. No evidence of supracrustal recycling in Si-O isotopes of Earth's oldest rocks 4 Ga ago. *Science Advances* **9**, eadf0693 (2023).
43. Hartnady, M.I.H. & Kirkland, C.L. A gradual transition to plate tectonics on Earth between 3.2 to 2.7 billion years ago. *Terra Nova* **31**, 129-134 (2019).
44. Liu, J. et al. Origin of the ca. 3.1 Ga Luanjiajie rock assemblage in the northeastern margin of the North China Craton: New constraints on the Mesoarchean geodynamic regime. *Precambrian Research* **376**, 106697 (2022).
45. Guo, B., Liu, S., Zhang, J. & Yan, M. Zircon U–Pb–Hf isotope systematics and geochemistry of Helong granite-greenstone belt in Southern Jilin Province, China: Implications for Neoproterozoic crustal evolution of the northeastern margin of North China Craton. *Precambrian Research* **271**, 254-277 (2015).
46. Diwu, C. et al. New evidence for ~4.45Ga terrestrial crust from zircon xenocrysts in Ordovician ignimbrite in the North Qinling Orogenic Belt, China. *Gondwana Research* **23**, 1484-1490 (2013).
47. Ma, Q. et al. Eoproterozoic to Paleoproterozoic crustal evolution in the North China Craton: Evidence from U-Pb and Hf-O isotopes of zircons from deep-crustal xenoliths. *Geochimica et Cosmochimica Acta* **278**, 94-109 (2020).
48. Reimink, J.R., Chacko, T., Stern, R.A. & Heaman, L.M. Earth's earliest evolved crust generated in an Iceland-like setting. *Nature Geoscience* **7**, 529-533 (2014).
49. O'Neil, J., Carlson, R.W., Francis, D. & Stevenson, R.K. Neodymium-142 evidence for Hadean mafic crust. *Science* **321**, 1828-1831 (2008).

50. Bauer, A.M., Fisher, C.M., Vervoort, J.D. & Bowring, S.A. Coupled zircon Lu–Hf and U–Pb isotopic analyses of the oldest terrestrial crust, the >4.03 Ga Acasta Gneiss Complex. *Earth and Planetary Science Letters* **458**, 37-48 (2017).
51. Kemp, A.I.S. et al. Hadean crustal evolution revisited: New constraints from Pb–Hf isotope systematics of the Jack Hills zircons. *Earth and Planetary Science Letters* **296**, 45-56 (2010).
52. Oliveira, E.P., McNaughton, N.J., Zincone, S.A. & Talavera, C. Birthplace of the São Francisco Craton, Brazil: Evidence from 3.60 to 3.64 Ga Gneisses of the Mairi Gneiss Complex. *Terra Nova* **32**, 281-289 (2020).
53. Wan, B. et al. Seismological evidence for the earliest global subduction network at 2 Ga ago. *Science Advances* **6**, eabc5491 (2020).

REVIEWERS' COMMENTS

Reviewer #1 (Remarks to the Author):

The authors have done a great job of improving the supplementary materials, and I think it is now much easier for readers to evaluate the quality of their data, which appears sound.

I also think the additional discussion of ^{142}Nd , ^{143}Nd isotope data strengthens the argument that the rocks in the NCC were derived by reprocessing crust that originally fractionated from the mantle in the Hadean to very early Archean.

I only have a few minor additional comments outlined below.

Specific comments:

Line 55: the 'thus' in this sentence feels out of place. I would remove it.

Line 66: craton growth and maturation processes?

Line 106-107: I struggle to follow this. The eHf is an empirical measurement and its uncertainty is controlled by the uncertainty in the $^{176}\text{Hf}/^{177}\text{Hf}$ measurement, uncertainty in the age (which affects the correction to initial compositions) and the uncertainty in the CHUR model used for normalisation. Do you mean to say that while these unradiogenic zircon indicate that some crust-mantle differentiation must have occurred in the Hadean to early Archean, the inferred source ages for these unradiogenic zircon varies depending on the assumed degree of mantle depletion?

Line 144-146: Similar secular eHf patterns are observed in other cratons (e.g. Mulder et al., 2020) so it might be worth mentioning that here.

Line 268-270: Might be worth mentioning mantle processes here, and brining in the stabilisation of thick subcontinental lithospheric mantle roots?

Line 301-302: I think this is debatable. Zircon oxygen can track melts generated from weathered rock, which probably requires some kind of crustal thickening / burial process, but I think linking it to subduction isn't necessarily straightforward. In lines 320-322 you appear to be more circumspect about this link. Maybe better to say 'has been proposed to track subduction...'

Reviewer #2 (Remarks to the Author):

Most of my concerns have been well addressed in the revision. I am OK with it.

Just a suggestion for Line 176: I suggest to revise "Crustal architecture of northeastern North China and

affinities among nuclei within the craton” as “Crustal architecture of northeastern North China and its affinity with other nuclei of the NCC”.

Comments in black

Responses in blue

Line numbers refer to the “clean” version of the manuscript *without* tracked changes.

Reviewers' comments:

Reviewer #1 (Remarks to the Author):

Reviewer #1 (Remarks to the Author):

The authors have done a great job of improving the supplementary materials, and I think it is now much easier for readers to evaluate the quality of their data, which appears sound.

I also think the additional discussion of $^{142,143}\text{Nd}$ isotope data strengthens the argument that the rocks in the NCC were derived by reprocessing crust that originally fractionated from the mantle in the Hadean to very early Archean.

I only have a few minor additional comments outlined below.

Response: We sincerely thank the reviewer for this comprehensive assessment.

Comment 1:

Line 55: the ‘thus’ in this sentence feels out of place. I would remove it.

Response: Thanks for your suggestion. We have removed ‘thus’.

Comment 2:

Line 66: craton growth and maturation processes?

Response: According to your suggestion, we have modified this sentence. Please refer to Line 68: “To establish craton growth and maturation processes of the Baishan nucleus”.

Comment 3:

Line 106-107: I struggle to follow this. The ϵHf is an empirical measurement and its uncertainty is controlled by the uncertainty in the $^{176}\text{Hf}/^{177}\text{Hf}$ measurement, uncertainty in the age (which affects the correction to initial compositions) and the uncertainty in the CHUR model used for normalisation. Do you mean to say that while these unradiogenic zircon indicate that some crust-mantle differentiation must have occurred in the Hadean to early Archean, the inferred source ages for these unradiogenic zircon varies depending on the assumed degree of mantle depletion?

Response: We appreciate the reviewer for clarifying this ambiguous expression. We agree with your suggestion. Please refer to Line 108-111: “These unradiogenic Hf isotopes indicate that some crust-mantle differentiation must have occurred in the Hadean to early Archaean, and the inferred source ages for these unradiogenic zircons vary depending on the assumed degree of mantle depletion.”

Comment 4:

Line 144-146: Similar secular eHf patterns are observed in other cratons (e.g. Mulder et al., 2020) so it might be worth mentioning that here.

Response: We appreciate this suggestion, and we agree with the reviewer that it is worth mentioning the work of Mulder et al. (2021) here. Please refer to Lines 151-152: "Similar Archaean crusts with secular evolved Hf patterns are also developed in other cratons, such as the Yilgarn and Slave¹."

Comment 5:

Line 268-270: Might be worth mentioning mantle processes here, and bringing in the stabilisation of thick subcontinental lithospheric mantle roots?

Response: Thanks for this good suggestion. We have made some modifications according to your suggestion. Please refer to Lines 276-277: "Such mantle processes also resulted in the stabilization of thick subcontinental lithospheric mantle roots. "

Comment 6:

Line 301-302: I think this is debatable. Zircon oxygen can track melts generated from weathered rock, which probably requires some kind of crustal thickening / burial process, but I think linking it to subduction isn't necessarily straightforward. In lines 320-322 you appear to be more circumspect about this link. Maybe better to say 'has been proposed to track subduction...'

Response: We agree with your suggestion, and replace "Zircon oxygen isotope analysis has been proven to be an effective method for tracing subduction zone processes" by "Zircon oxygen isotope analysis has been proposed to track subduction zone processes". Please refer to Lines 312-313.

Reviewer #2 (Remarks to the Author):

Most of my concerns have been well addressed in the revision. I am OK with it. Just a suggestion for Line 176: I suggest to revise "Crustal architecture of northeastern North China and affinities among nuclei within the craton" as "Crustal architecture of northeastern North China and its affinity with other nuclei of the NCC".

Response: We sincerely thank the reviewer for this comprehensive assessment. We have changed the subtitle "Crustal architecture of northeastern North China and affinities among nuclei within the craton" into "Crustal architecture of northeastern North China and its affinity with other nuclei of the North China Craton". Please refer to Lines 183-184.

Reference

1. Mulder JA, Nebel O, Gardiner NJ, Cawood PA, Wainwright AN, Ivanic TJ. Crustal rejuvenation stabilised Earth's first cratons. *Nat Commun* **12**, 3535 (2021).